# Fine-Grained Activation Steering: Steering Less, Achieving More

**Zijian Feng**[1]     **Tianjiao Li**[1]     **Zixiao Zhu**[1]     **Hanzhang Zhou**[1]     **Junlang Qian**[1]
**Li Zhang**[1]     **Jia Jim Deryl Chua**[2]     **Lee Onn Mak**[2]     **Gee Wah Ng**[2]     **Kezhi Mao**[1,*]

[1]School of Electrical and Electronic Engineering, Nanyang Technological University, Singapore
[2]Home Team Science and Technology Agency (HTX), Singapore
{feng0119, zixiao001, hanzhang001, junlang001,
zhan0735}@e.ntu.edu.sg
{tianjiao.li, ekzmao}@ntu.edu.sg
{deryl_chua, mak_lee_onn, ng_gee_wah}@htx.gov.sg

## Abstract

Activation steering has emerged as a cost-effective paradigm for modifying large language model (LLM) behaviors. Existing methods typically intervene at the block level, steering the bundled activations of selected attention heads, feedforward networks, or residual streams. However, we reveal that block-level activations are inherently heterogeneous, entangling beneficial, irrelevant, and harmful features, thereby rendering block-level steering coarse, inefficient, and intrusive. To investigate the root cause, we decompose block activations into fine-grained atomic unit (AU)–level activations, where each AU-level activation corresponds to a single dimension of the block activation, and each AU denotes a slice of the block weight matrix. Steering an AU-level activation is thus equivalent to steering its associated AU. Our theoretical and empirical analysis show that heterogeneity arises because different AUs or dimensions control distinct token distributions in LLM outputs. Hence, block-level steering inevitably moves helpful and harmful token directions together, which reduces efficiency. Restricting intervention to beneficial AUs yields more precise and effective steering. Building on this insight, we propose AUSteer, a simple and efficient method that operates at a finer granularity of the AU level. AUSteer first identifies discriminative AUs globally by computing activation momenta on contrastive samples. It then assigns adaptive steering strengths tailored to diverse inputs and selected AU activations. Comprehensive experiments on multiple LLMs and tasks show that AUSteer consistently surpasses advanced baselines while steering considerably fewer activations, demonstrating that *steering less achieves more* [1].

## 1 Introduction

In the era of large language models (LLMs), activation steering has emerged as a powerful paradigm for modulating model behavior on downstream tasks (Zou et al., 2023; Li et al., 2023b; Rimsky et al., 2024). Unlike reinforcement learning from human feedback (Bai et al., 2022), supervised fine-tuning (Wei et al., 2022), or prompt engineering (Brown et al., 2020), activation steering intervenes directly in the LLM intermediate activations during forward propagation, enabling fine-grained control without additional training. Prior work (Turner et al., 2023; Rimsky et al., 2024; Han et al., 2024; Wang et al., 2025a;b;c) generally builds task-specific steering vectors and injects them at inference time as biases or rescaling factors in selected LLM components, thereby steering the model toward the target objective.

---

* Corresponding author.
[1]Code: https://github.com/zijian678/AUSteer

However, a common practice in existing methods is **block-level steering**, where a "block" denotes the multi-head attention (MHA), the feed-forward network (FFN), or the layer's residual stream. As shown in Figure 1 (a), the intervention is vector-level: every dimension of the selected block's activation is bundled and steered simultaneously. One of the main limitations of block-level intervention is that it ignores **heterogeneity** within block activations. These activations often span hundreds or thousands of dimensions, each indicating a different feature. Some features are beneficial for the task, while others are irrelevant or harmful. As a result, block level steering is (1) too coarse: a block can be decomposed into finer functional units, and treating it as a single entity prevents precise targeting; (2) inefficient: steering the entire block amplifies both useful and harmful signals, which reduces efficiency and risks performance degradation; and (3) overly intrusive: it modifies many dimensions unnecessarily, increasing the intervention footprint.

In greater depth, we empirically and theoretically justify the heterogeneity of block-level activations. We first decompose block-level activations into finer-grained atomic unit (AU) activations, where each AU-level activation corresponds to a single dimension of the block activation, and each AU denotes a slice of the block weight matrix. Steering an AU-level activation is thus equivalent to steering its associated AU. As shown in Figure 1 (b), each AU-level intervention targets a single dimension[2]. Both the intervention value and the affected activation are scalars. Empirically, we find that AU-level steering effects vary widely: some dimensions improve performance, some degrade it, and others are neutral, confirming heterogeneity. In many cases, steering a single dimension or a small subset outperforms steering the entire block.

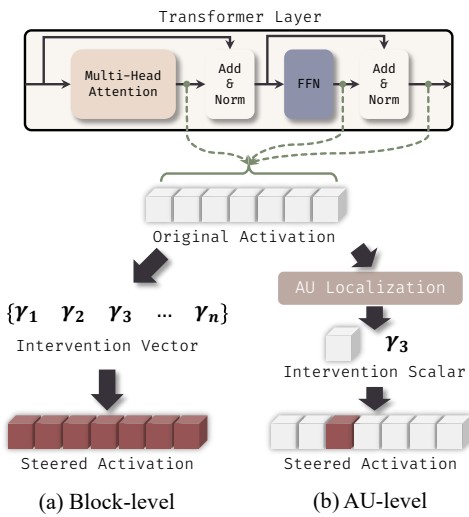

Figure 1: Comparison of block-level steering (prior work) and AU-level steering (Ours).

Our theoretical analysis reveals that the **heterogeneity stems from different AUs modulating distinct output-token distributions**. Steering a single dimension therefore shifts the model's output distribution toward the distribution controlled by that AU. Some AUs favor task-irrelevant or harmful tokens; steering their dimensions degrades performance. This also explains why block-level steering, which mixes helpful and harmful AUs together, can underperform more targeted and precise AU-level steering. Targeting only beneficial AUs can reduce the intervention footprint and improve efficiency, that is, **steering less achieves more**.

Beyond the promise of AU-level steering, these findings also pose challenges: (1) how can we localize the most important AUs for intervention? and (2) how can we ensure adaptive steering across diverse inputs and AUs?

To address these challenges, we introduce **AUSteer**, a simple and efficient method with two components. First, we propose **activation momentum**, a new metric that analyzes each activation's momentum in positive and negative samples to evaluate its discriminative power. This counting-based metric supports global comparison and avoids the issue of increasing activation magnitudes across layer. We then localize the most discriminative AUs or activations for steering. Second, to ensure **adaptivity** across inputs and AUs, we assign a per sample steering scalar that follows the original activation pattern rather than a constant shift. This makes the update scale with the current activation, and preserves direction. We also assign dynamic steering strength to each AU according to its discriminative power, with important AUs receiving higher strength. We compare AUSteer with state-of-the-art (SOTA) methods that intervene at the block level by steering hundreds to thousands of activations. Using far fewer steered activations (at most 100), AUSteer significantly outperforms these methods across diverse tasks, demonstrating that **steering less achieves more**.

---

[2]For clarity: a block-level activation is a vector associated with a block (MHA, FFN, or a layer's residual stream), usually comprising hundreds to thousands of dimensions, whereas an AU-level activation is a scalar corresponding to a single dimension within that block activation.

The contributions of this work are summarized as follows:

- Conceptually, we study the heterogeneity within block-level activations and its root causes, both theoretically and empirically, and propose decomposing block-level intervention into fine-grained AU-level intervention (§3).

- Methodologically, we propose AUSteer, a framework that localizes discriminative AUs with activation momenta for steering, and ensures adaptivity across diverse inputs and AUs (§4).

- Empirically, we evaluate AUSteer on multiple LLMs of varying sizes across diverse tasks, including commonsense reasoning, mathematical problem solving, and open-ended generation. With less intrusive intervention, AUSteer significantly outperforms other SOTA activation steering methods, underscoring that steering less achieves more (§5).

## 2 Related Work

Activation steering (also known as activation editing) has become a popular and cost-effective approach for modifying LLM behaviors and aligning them with downstream tasks (Turner et al., 2023; Rimsky et al., 2024; Han et al., 2024; Wang et al., 2025c; Soo et al., 2025; Stickland et al., 2024; Li et al., 2023c; Wang et al., 2025a;b; Stolfo et al., 2025). The standard workflow involves extracting steering vectors from prompts or contrastive samples and injecting them into LLMs at inference time. Most of these methods intervene at the **block level**. For instance, at **MHA blocks**, ITI (Li et al., 2023b) derives steering vectors from contrastive activations in attention blocks and then applies interventions using the extracted vectors to important heads. Bhattacharjee et al. (2024) compute category-specific activations from attention heads to reduce unsafe responses. In **residual streams**, CAA (Rimsky et al., 2024) extracts vectors from positive and negative samples and applies them to residual streams, while van der Weij et al. (2024) extend this approach to multi-vector steering across residual streams. EAST (Rahn et al., 2024) obtains steering vectors by weighting input prompts with entropy and injects them into the layer outputs. Postmus & Abreu (2024) use multiple steering vectors as a conceptor to redirect behaviors via residual stream activations. Safety methods such as SafeSwitch (Han et al., 2025) and Safety Arithmetic (Hazra et al., 2024) intervene in residual streams to suppress harmful outputs. Konen et al. (2024) extract steering vectors from layer outputs to control emotion and writing style, while AnyEdit (Jiang et al., 2025) updates hidden states and knowledge by steering layer outputs. More recently, Stolfo et al. (2025) steer residual streams to enhance instruction following. Some methods can operate across multiple blocks. For example, SADI (Wang et al., 2025b) computes steering vectors from **MHA**, **FFN**, or **residual streams**, then applies mask-adaptive steering.

Notably, STA (Wang et al., 2025a) identifies atoms in pretrained sparse autoencoders (SAEs) (Lieberum et al., 2024; He et al., 2024; Gao et al., 2025) of target LLMs and steers *residual streams* using these localized units. Although STA uses the term *atom*, its meaning differs from ours: in STA, an atom is a knowledge unit in an SAE, whereas in our work an atom is a unit in the original LLM weight matrices. Methodologically, STA depends on pretrained SAEs that currently exist for only a few model families such as LLaMA3.1 (Touvron et al., 2023) and Gemma2 (Team et al., 2024), which limits generalization. Moreover, STA's intervention remains at the block level as the computed vectors are injected into the residual stream.

## 3 Heterogeneous Block Activations: Steering Less Achieves More

### 3.1 Block Decomposition

We first show how computations within LLM blocks can be decomposed into fine-grained AU calculations. The backbone architecture of LLMs is the Transformer, which consists of attention blocks and FFN blocks in every layer. The outputs of these blocks are added to the layer residual stream for forward propagation. In both MHA and FFN, weight matrix computations ($Q, K, V, O$ in MHA and the up projection and down projection in FFN) are linear projections of the form $\mathbf{y} = \mathbf{W}\mathbf{x}$, where $\mathbf{x}$ is the input activation, $\mathbf{W}$ is the weight matrix, and $\mathbf{y}$ is the output activation.

In existing studies, block activations ($\mathbf{x}$ and $\mathbf{y}$) are typically treated as indivisible vectors. Steering vectors are calculated and applied at this coarse block level. To decompose blocks into finer-grained units, we reformulate the linear projection as

$$\mathbf{y} = \mathbf{W}\mathbf{x} = \sum_i x_i \, \mathbf{W}_{:,i}. \tag{1}$$

Here, $x_i$ denotes the $i$-th dimension of the input activation $\mathbf{x}$. This formulation allows us to isolate each single-dimensional activation. In this view, every scalar $x_i$ serves as the coefficient for the corresponding column $\mathbf{W}_{:,i}$ of the weight matrix. We refer to each column $\mathbf{W}_{:,i}$ as an **Atomic Unit** (AU) in our study.[3] In this way, steering the $i$-th dimension activation $x_i$ is equivalent to steering the corresponding $i$-th AU. To clarify:

- $\mathbf{x}, \mathbf{y}$: block-level activations, represented as vectors (the standard formulation in prior work).
- $\mathbf{W}_{:,i}$: the $i$-th column of the weight matrix $\mathbf{W}$, representing the $i$-th AU.
- $x_i$: $i$-th dimension or $i$-th AU-level activation, which is a scalar and the coefficient for the $i$-th AU.

### 3.2 HETEROGENEITY IN BLOCK ACTIVATIONS

In this section, we examine the heterogeneous effects of AU-level activations within the block activation. To ensure generalizability, we adopt two representative steering methods: the pioneering ITI (Li et al., 2023b) and SOTA SADI (Wang et al., 2025b), applying them to MHA and FFN blocks. We use LLaMA2-7B-Chat (Touvron et al., 2023) as the backbone model and BoolQ (Clark et al., 2019) as the illustrative dataset, where the model answers "yes" or "no" for each question and the accuracy is reported. The experimental setup follows SADI, as described in Appendix B.

We first use ITI and SADI to identify important attention heads and FFNs for intervention, then compare six conditions: (1) **Baseline**, the original model without steering; (2) **ITI**, block-level intervention on attention head activations; (3) **SADI** (Wang et al., 2025b), block-level steering on attention heads (128 dimensions) and FFNs (4096 dimensions); (4) **Dimension Sweep**, steering single dimensions rather than whole blocks, sampling one of every four dimensions in attention heads and one of every 100 in FFNs; (5) **Positive Combination**, steering a small subset of beneficial dimensions; and (6) **Mixed Comb.**, steering a subset of beneficial and detrimental dimensions .

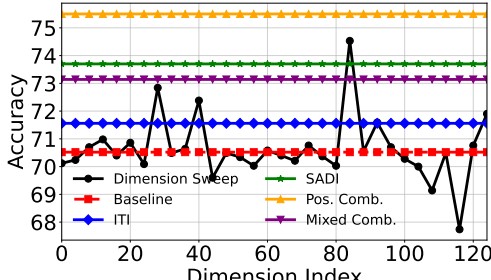 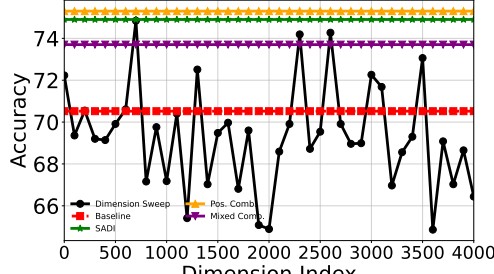

(a) Steering results for the 7th attention output in layer 27. **Positive Combination**: steering four beneficial dimensions (28, 40, 84, 92). **Mixed Combination**: steering those four plus two detrimental dimensions (108, 116).

(b) Steering results for the FFN output in layer 20. **Positive Combination**: steering four beneficial dimensions (0, 1300, 2300, 3000). **Mixed Combination**: steering those four plus two detrimental dimensions (1200, 1700).

Figure 2: Heterogeneous steering results for MHA and FFNs.

Figure 2a shows the results of interventions on the 7th attention head at the 27th layer. The original model achieves 70.52% accuracy, ITI reaches 71.56%, and SADI achieves 73.70%. Steering individual AU activations, however, produces highly **heterogeneous** outcomes: some dimensions

---

[3]Each column of $\mathbf{W}$ corresponds to an AU, while each row corresponds to what is traditionally termed a "neuron." To ensure rigor and avoid confusion, we adopt the term AU rather than neuron.

degrade performance, while others improve it. Notably, steering a single dimension can outperform full block steering. For example, steering the 84th dimension alone achieves 74.53%, surpassing the baseline, ITI, and SADI. Furthermore, steering only four positively contributing dimensions (Pos. Comb.) yields even stronger results. While introducing detrimental dimensions (Mixed Comb.), the perform drops. Similar observations hold for FFN blocks in Figure 2b. Additional empirical results for other attention heads and FFNs are provided in Appendix A. These findings indicate that block-level steering is inefficient, as it mixes beneficial and detrimental components. In contrast, fine-grained AU-level steering enables selective amplification of useful features, achieving more effective control. In short, **steering less achieves more**.

### 3.3 INTERPRETING THE HETEROGENEITY

To explain the observed heterogeneity, as discussed above, we treat the block activation as coefficients on an AU basis, so steering a single dimensional activation $x_i$ is equivalent to steering its AU. Building on prior theory of interpreting LLMs in the embedding space (Geva et al., 2022; Dar et al., 2023), different AUs may control different token distributions in LLM outputs. Steering task-relevant AUs promotes the probability of task-specific tokens, whereas steering task-irrelevant AUs may increase the probability of uninformative or even harmful tokens. This provides a theoretical justification for the observed heterogeneity.

To further validate this, we first examine the **convergence** behavior of AU steering: different AUs govern different output token distributions, and as steering strength increases, the LLM's output tends to converge to the AU's token distribution. For the selected 7th attention head at the 27th layer, we scale the AU coefficient from 10 to an extremely large value (100,000) and compute the normalized KL divergence between the output at each strength and the output at 100,000. In Figure 3, columns 1 and 2 show these divergences for the 44th AU and the 84th AU. The divergence decreases with strength, indicat-

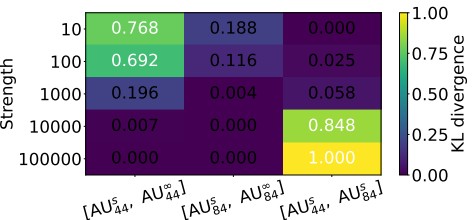

Figure 3: Pairwise KL divergence when steering different AUs. $s$ means strength.

ing convergence. Column 3 shows the pairwise KL divergence between the 44th AU and the 84th AU across strengths. The divergence increases with strength, indicating that the two AUs tend to drive the model toward different output distributions.

Figure 4 illustrates this phenomenon by reporting the top-5 output tokens after steering three different AUs with single-dimensional activations. The input prompt is a question from BoolQ dataset with the answer "yes". Steering $x_{84}$ promotes the correct answer token "yes" while suppressing the incorrect "no", thereby improving accuracy. In contrast, steering dimensions $x_{44}$ or $x_{100}$ elevates task-irrelevant or incorrect tokens, resulting in degraded performance. These observations align with the accuracies shown in Figure 2a.

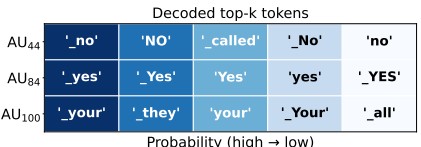

Figure 4: Top-k decode tokens controlled by different AUs. The answer to input prompt is "yes".

In summary, heterogeneity arises because each AU governs a distinct output-token distribution. Block-level activations inevitably mix beneficial, irrelevant, and harmful AUs, making block-level interventions coarse, inefficient, and intrusive. By contrast, selectively steering only the helpful AUs amplifies the desired distribution and enables more efficient control.

## 4 METHODOLOGY: AUSTEER

Breaking block-level interventions into finer-grained AU-level interventions has shown promise for modifying LLM behaviors. Yet AU-level steering faces some fundamental challenges: identifying important AU-level activations for intervention and ensuring adaptability across diverse inputs and AUs. To address these challenges, we propose AUSteer shown as Figure 5.

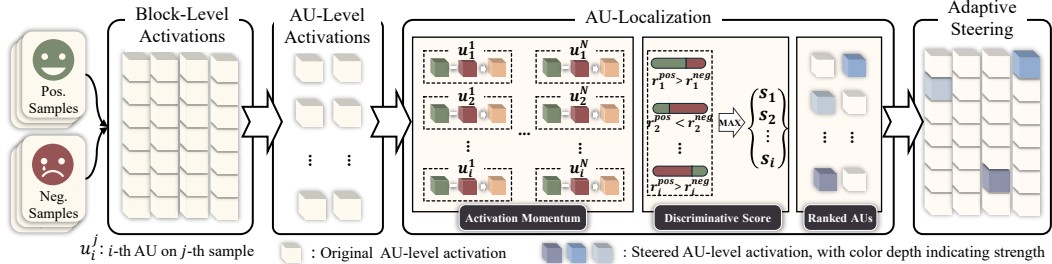

Figure 5: Overview of AUSteer: (1) AU localization using activation momentum and discriminative scores; and (2) Adaptive steering across diverse inputs and AUs.

## 4.1 ATOMIC UNIT LOCALIZATION

The first challenge is to identify which AUs and their activations should be steered. Prior work often uses probing (Li et al., 2023b) or activation values (Wang et al., 2025b) of contrastive pairs as importance metrics. Here, a contrastive pair consists of a positive example with a correct or high-quality response and a matched negative example with an incorrect or low-quality response. However, probing requires additional training resources and does not transfer well to single-dimension settings of AUs, while activation magnitudes tend to increase with layer depth, making cross-layer comparisons unreliable. To overcome these limitations, we propose an **activation momentum** strategy.

Given $N$ pairs of contrastive samples[4], an AU is **discriminative** if its activation coefficient $x_i$ consistently separates positives from their matched negatives. Concretely, if $x_i$ is systematically higher (or lower) for the positive sample of each pair than for the negative sample, the AU promotes (or suppresses) activation for positives relative to negatives. Such consistency indicates that the AU distinguishes positive from negative cases and is therefore task-relevant.

Formally, let $u_i$ denote the $i$-th AU with the activation $x_i$. For the $j$-th sample pair, we define the activation momentum as

$$m_i^j = x_i^{j,\text{pos}} - x_i^{j,\text{neg}},$$

where $x_i^{j,\text{pos}}$ and $x_i^{j,\text{neg}}$ are the activation values of $u_i$ on the $j$-th positive and negative sample, respectively. Note that both $x_i^{j,\text{pos}}$ and $x_i^{j,\text{neg}}$ are one-dimensional scalars as defined in Eq.1. When $m_i^j > 0$, the AU exhibits an activation promotion effect for positive samples, whereas $m_i^j < 0$ indicates a suppression effect. By counting the occurrences of promotion and suppression across samples, we can assess whether an AU shows a consistent effect on positive or negative cases, thereby quantifying its discriminative power. The proportions of positive and negative momenta are then given by

$$r_i^{\text{pos}} = \frac{1}{N} \sum_{j=1}^{N} \mathbb{1}(m_i^j > 0), \qquad r_i^{\text{neg}} = \frac{1}{N} \sum_{j=1}^{N} \mathbb{1}(m_i^j < 0). \qquad (2)$$

The discriminative score of the $i$-th AU is defined as:

$$s_i = \max(r_i^{\text{pos}}, r_i^{\text{neg}}).$$

This scoring provides a unified scale for cross-layer comparison, allowing us to rank AUs globally and select the most important $k$ AUs for steering. To verify how activation momentum contributes to discriminative causality and the final model outputs, we provide both theoretical and empirical analyses in Appendix H.

## 4.2 ADAPTIVE STEERING

The steering of an activation $x_i$ should be adaptive in two respects. First, it should adapt to diverse inputs. Different samples produce activations with different magnitudes and semantic contexts. Adding a constant vector ignores this variation, can distort useful directions, and may impair model

---

[4]Details of contrastive sample construction can be found in Appendix B.1

performance. We therefore obtain the steered activation by $\hat{x}_i = x_i + \gamma_i x_i$, which scales the current activation, preserves its sign, and adapts well across varies samples.

Second, steering should adapt across AUs. More discriminative AUs receive stronger steering, while less important ones receive weaker intervention. This concentrates changes on useful AUs and limits unnecessary perturbations. To achieve this, we compute $\gamma_i$ as

$$\gamma_i = \begin{cases} \alpha\, r_i^{\text{pos}}, & r_i^{\text{pos}} > r_i^{\text{neg}}, \\ -\alpha\, r_i^{\text{neg}}, & \text{otherwise}, \end{cases}$$

where $\alpha$ is a global steering strength factor, $r_i^{\text{pos}}$ and $r_i^{\text{neg}}$ are the positive and negative discriminative scores of the $i$-th AU calculated by Eq.2. The steering direction is determined by whether the AU has a promotive or suppressive effect. Finally, for the selected AUs, activations are updated as

$$\hat{x}_i = x_i + \gamma_i x_i.$$

**Applicability of AUSteer.** The proposed AUSteer can be applied to all key components of LLMs, including MHA, FFN, and residual streams, as the analysis above holds uniformly across these modules. Unlike previous approaches that operate on entire block-level activations, AUSteer intervenes only on the most important dimensions within each block activation. This yields interventions that are both more efficient and less intrusive, embodying the principle of **steering less to achieve more**.

## 5 EXPERIMENTS

### 5.1 EXPERIMENTAL SETTINGS

**Tasks and Evaluation Metrics.** We evaluate AUSteer on three types of tasks:

- **Commonsense reasoning.** We use widely adopted datasets including BoolQ (Clark et al., 2019), COPA (Gordon et al., 2012), and WinoGrande (Sakaguchi et al., 2021), and report **accuracy** of the model's responses using exact match.

- **Math problem solving.** We experiment with SVAMP (Patel et al., 2021) and MAWPS (Koncel-Kedziorski et al., 2016), where the model is required to solve math questions with or without reasoning. We evaluate **accuracy** by comparing the predicted answer with the correct number.

- **Open-ended generation.** We employ RealToxicPrompts (Gehman et al., 2020) and BPO (Cheng et al., 2024). For RealToxicPrompts, which contains challenging prompts that often elicit toxic content, we apply different steering methods to reduce toxicity. Automatic evaluation follows prior work (Wang et al., 2025a): **detoxification** performance, where toxicity is measured using the Perspective API [5]. For BPO, which aligns model outputs with human-preferred behaviors, we adopt the automatic evaluation protocol of Zheng et al. (2023); Liang et al. (2024) and report $\Delta\text{WR} = \text{WR}_{steered} - \text{WR}_{original}$. The win-rates are obtained by using GPT-5-mini (OpenAI, 2025) and prompts from Liang et al. (2024) to compare the original and steered responses. For both datasets, **human evaluation** is conducted, where 3 annotators assess text **quality** (fluency, diversity) and **alignment** with the target objective on a 1–5 scale.

**Target LLMs.** We evaluate AUSteer on a diverse set of LLMs: (1) LLaMA2-7B-Chat (Touvron et al., 2023), which serves as the backbone in many related studies; (2) Gemma2-9B-it (Team et al., 2024), a strong decoder-only model for text generation; and (3) Qwen3-8B (Yang et al., 2025), one of the most recent LLMs. To further assess scalability, we also experiment with other LLMs and larger variants (e.g., 13B, 27B), with results reported in §5.4.

**Baselines.** We compare AUSteer against several competitive activation steering methods:

- **ITI** (Li et al., 2023b), which uses contrastive samples to identify important attention heads, then derives steering vectors from activation differences for intervention.

- **CAA** (Rimsky et al., 2024), which extracts steering vectors from activation differences in residual streams and applies them at the block level.

---

[5]https://www.perspectiveapi.com

Table 1: Overall results of baseline methods and the proposed AUSteer across seven tasks. "#Acts" denotes the number of intervened activations for each method. $k_h$ indicates the number of selected attention heads, ranging from 2 to 64. Results with $^\dagger$ are from (Wang et al., 2025a;b).

| Model | Method | #Acts ($\downarrow$) | Commonsense Reasoning ($\uparrow$) | | | Math Problem Solving ($\uparrow$) | | Avg. | Open Generation ($\uparrow$) | |
| --- | --- | --- | --- | --- | --- | --- | --- | --- | --- | --- |
| | | | BoolQ | COPA | WinoG. | SVAMP | MAWPS | Acc. | Detox. | $\triangle$WR |
| LLaMA2-7B-Chat | Vanilla | 0 | $70.52^\dagger$ | $70.80^\dagger$ | $50.91^\dagger$ | 36.00 | 51.83 | 56.01 | – | – |
| | ITI | $k_h \cdot 128$ | $74.10^\dagger$ | $77.20^\dagger$ | $52.80^\dagger$ | 36.67 | 54.08 | 58.97 | 83.49 | 12.50 |
| | CAA | 4096 | $74.98^\dagger$ | $75.20^\dagger$ | $52.64^\dagger$ | 34.33 | 55.21 | 58.47 | 84.57 | 11.00 |
| | SADI | $k_h \cdot 128$ | $74.35^\dagger$ | $78.80^\dagger$ | $53.04^\dagger$ | 36.33 | 54.93 | 59.49 | 86.32 | 13.50 |
| | AUSteer-Head | $\leq 100$ | **76.27** | 75.40 | 52.80 | **37.67** | 56.06 | 59.64 | **89.66** | 16.00 |
| | AUSteer-FFN | $\leq 100$ | 75.57 | **82.80** | **53.28** | 37.00 | **58.03** | **61.34** | 89.24 | **22.00** |
| Gemma2-9B-it | Vanilla | 0 | 86.64 | 77.40 | 67.25 | 67.67 | 92.11 | 78.21 | – | – |
| | ITI | $k_h \cdot 256$ | 86.82 | 88.40 | 68.11 | 69.00 | 93.52 | 81.17 | 98.83 | 23.00 |
| | CAA | 3584 | 86.85 | 89.40 | 68.35 | 68.00 | 92.96 | 81.11 | $98.75^\dagger$ | 23.50 |
| | SADI | $k_h \cdot 256$ | 86.79 | 92.40 | 69.53 | 68.00 | 93.52 | 82.05 | 99.08 | 25.50 |
| | STA | 3584 | 87.03 | 91.60 | 69.61 | 68.67 | 92.39 | 81.86 | $99.33^\dagger$ | 25.00 |
| | AUSteer-Head | $\leq 100$ | 86.79 | 91.00 | 70.72 | **71.00** | **94.65** | 82.83 | 99.00 | 26.50 |
| | AUSteer-FFN | $\leq 100$ | **87.25** | **97.60** | **70.88** | 70.00 | 94.08 | **83.96** | 99.25 | **30.00** |
| Qwen3-8B | Vanilla | 0 | 87.43 | 89.00 | 63.61 | 68.67 | 84.51 | 78.64 | – | – |
| | ITI | $k_h \cdot 128$ | 87.40 | 90.80 | 65.59 | 69.67 | 87.32 | 80.16 | 75.15 | 26.00 |
| | CAA | 4096 | 87.83 | 91.20 | 64.01 | 71.67 | 88.45 | 80.63 | 76.23 | 27.00 |
| | SADI | $k_h \cdot 128$ | 87.65 | 93.20 | 64.25 | 73.00 | 85.92 | 80.80 | 78.07 | 25.50 |
| | AUSteer-Head | $\leq 100$ | 87.71 | **95.40** | 65.35 | **76.33** | **91.27** | **83.21** | 80.90 | 26.00 |
| | AUSteer-FFN | $\leq 100$ | **88.20** | 90.80 | **67.25** | 71.67 | 89.58 | 81.50 | **81.65** | **34.00** |

- **SADI** (Wang et al., 2025b), which localizes important attention heads, FFNs, or layers via activation differences, and applies adaptive steering through masking and scaling. We report results for its best-performing variant, SADI-HEAD.

- **STA** (Wang et al., 2025a), which identifies important atoms in sparse autoencoders (SAEs) of the target LLM, then applies steering vectors to residual streams. Since pretrained SAEs are currently available only for LLaMA 3.1 and Gemma2, we report its results only on Gemma2.

**AUSteer Variants.** The proposed method can be applied to any key component of LLMs. Since the two core modules in each Transformer layer are the attention and FFN blocks, we validate the generalizability of AUSteer by implementing two variants: **AUSteer-Head**, which steers AU-level activations in MHA, and **AUSteer-FFN**, which steers AU-level activations in FFN.

**Implementation details** for AUSteer and all other baseline methods, including contrastive pair construction, dataset statistics, and prompt templates, are provided in Appendix B.

**Hyperparameter settings.** For baseline methods, we perform hyperparameter sweeps following the recommendations in their papers to ensure a fair comparison. AUSteer introduces two hyperparameters: (1) $k$, the number of AU-level activations selected for steering; and (2) $\alpha$, a global steering-strength factor. To verify the claim that we can *steer less to achieve more*, we cap the number of steered activations at 100 and then run the sweep. Full details appear in Appendix C.

## 5.2 MAIN RESULTS

**AUSteer significantly improves commonsense reasoning and math problem solving with minimal intervention.** Table 1 reports overall results on LLaMA2-7B-Chat, Gemma2-9B-it, and Qwen3-8B. Across all five tasks on commonsense reasoning and math questions, either AUSteer-FFN or AUSteer-Head attains the highest average accuracy while steering at most 100 activations, in contrast to SADI's $k_h \times 128$ head interventions and CAA/STA, which modify thousands of activations. Concretely, AUSteer-FFN improves the average over SADI by **+1.85** on LLaMA2-7B-Chat (61.34 vs. 59.49), **+1.91** on Gemma2-9B-it (83.96 vs. 82.05), and **+0.7** on Qwen3-8B. AUSteer-Head is also competitive, exceeding SADI on Qwen3-8B by **+2.41** under the same low-budget constraint. Beyond averages, AUSteer-Head or AUSteer-FFN consistently achieves the best scores on individual tasks across the five commonsense and math benchmarks.

**AUSteer improves open-ended generation.** In automatic evaluation (Table 1), AUSteer significantly increases detoxification rates under toxic prompts. Compared with SADI, it yields around 2%-3% higher detoxification on Llama2 and Qwen3.

Table 2: Human evaluation on open-ended generation tasks.

| | LLaMA2-7B-Chat | | Gemma2-9B-it | | | Qwen3-8B | |
|---|---|---|---|---|---|---|---|
| | SADI | AUSteer | SADI | STA | AUSteer | SADI | AUSteer |
| Quality (↑) | 3.3 | **3.4** | 4.2 | **4.4** | 4.3 | 4.1 | **4.3** |
| Alignment (↑) | 3.6 | **3.8** | 4.5 | **4.7** | **4.7** | 3.9 | **4.1** |

On BPO datasets, AUSteer steers models toward human-preferred responses, improving win-rates (ΔWR) by 8.5%, 4.5%, and 7% on the three LLMs, respectively. In human evaluation (Table 2), AUSteer outperforms baselines in most cases on generation quality (fluency and diversity) and on alignment with the generation target.

## 5.3 ABLATION STUDIES

We evaluate the contribution of each component in AUSteer: AU localization and adaptive steering. To assess the proposed activation momentum localization, we compare it with (1) **random localization**, which selects activations at random for steering, and (2) **activation difference** across contrastive samples for localization, as introduced in SADI. To assess adaptive steering, we compare it with (3) a **fixed steering vector**, which replaces $\gamma_i x_i$ with the mean activation difference, following ITI, and (4) a **fixed steering strength** $\gamma$, which applies the same strength across all selected AUs.

Table 3: Ablation study results on Gemma2-9B-it.

| Method | Avg. Acc |
|---|---|
| AUSteer-FFN | 83.96 |
| Random Loc. | 79.08 (**-4.88**) |
| Act. Diff. | 83.12 (**-0.84**) |
| Fixed. Vec. | 82.05 (**-1.91**) |
| Fixed. Strength | 83.04 (**-0.92**) |

The average accuracy across commonsense reasoning and math questions are shown in Table 3. When using random or activation-difference localization, steering performance drops substantially, verifying the effectiveness of the proposed activation momentum-based localization. Similarly, replacing adaptive steering with a fixed vector or fixed strength reduces performance by 1.91 and 0.92, respectively, demonstrating the importance of adaptivity across diverse inputs and AUs.

## 5.4 SCALABILITY AND GENERALIZABILITY OF AUSTEER

Table 4: Experimental results on more LLMs.

| | LLaMA3.1-8B-Instruct | | | LLaMA2-13B-Chat | | | Gemma2-27B-it | | |
|---|---|---|---|---|---|---|---|---|---|
| | BoolQ | COPA | WinoG. | BoolQ | COPA | WinoG. | BoolQ | COPA | WinoG. |
| Vanilla | 82.57 | 83.80 | 57.77 | 84.01 | 89.00 | 53.99 | 86.88 | 86.00 | 63.61 |
| AUSteer-Head | 83.18 | 86.00 | 60.38 | 85.25 | 91.00 | 59.43 | 88.10 | 90.20 | 67.25 |
| AUSteer-FFN | 83.79 | 86.00 | 61.56 | 85.02 | 91.20 | 58.88 | 88.41 | 89.80 | 66.30 |

We evaluate AUSTEER on larger and varied LLMs, including LLaMA3.1-8B-Instruct, LLaMA2-13B-Chat (Touvron et al., 2023), and Gemma2-27B-it (Team et al., 2024), on commonsense reasoning tasks. Table 4 reports the results. Both AUSTEER-HEAD and AUSTEER-FFN substantially improve the base models, confirming the method's scalability and generalizability. More results on larger LLMs with diverse structures including Qwen3-30B-A3B and Llama-3.3-70B-Instruct are provided in Appendidx G.

## 5.5 FURTHER ANALYSIS

To investigate the internal mechanisms of AUSteer more comprehensively, we provide the following discussions.

- Appendix C. We illustrate the hyperparameter sweep for $k$ and $\alpha$ and report their optimal values across tasks. We also provide guidelines for hyperparameter search in both resource-sufficient and resource-constrained settings.

- Appendix D. We characterize activation momentum for different AUs and analyze the locations of AUs within MHA and FFN.

- Appendix E. We present and discuss the overlap of localized AUs across tasks.

- Appendix F. We evaluate AUSTEER under varying numbers of contrastive pairs used for AU localization.

- Appendix G. We demonstrate AUSteer's scalability on larger LLMs with diverse architectures, including Qwen3-30B-A3B (a sparse MoE model) and Llama-3.3-70B-Instruct (evaluated in its 4-bit quantized form).

- Appendix H. We verify how activation momentum contributes to discriminative causality and final model outputs, providing both theoretical and empirical analyses.

- Appendix I. We present a detailed analysis of AUSteer's efficiency and computational overhead compared with baseline methods. Overhead results on Llama-3.3-70B-Instruct are also included.

- Appendix J. We experiment with additional control variants of AUSteer—such as steering all AUs or broader subsets—and confirm that, as with AUSteer, steering should be limited to task-relevant and beneficial AUs rather than blindly steering all or large numbers of units.

- Appendix K. To determine whether we should promote useful AUs or suppress unhelpful ones, we conduct both empirical and theoretical analyses and show that promotion consistently outperforms suppression.

## 6 CONCLUSION

In this work, we investigate the heterogeneity and its root cause of block-level activations and propose AUSteer, a fine-grained AU-level activation steering method. AUSteer localizes salient AUs via activation momentum and assigns dynamic steering strengths per input and AU. Extensive experiments show that, with far fewer intervened activations, AUSteer significantly outperforms state-of-the-art methods across diverse tasks, demonstrating that *steering less achieves more*.

## ACKNOWLEDGMENTS

We extend our heartfelt gratitude to the reviewers for their insightful and constructive feedback. This research was supported by the Home Team Science and Technology Agency (HTX), Singapore under the NTU-HTX collaboration project: *Parsimonious Domain Specific Large Language Model Enabled Multimodality Sensemaking*. We express our sincere appreciation to HTX for their continued support and collaboration.

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

## A    More Results on Activation Heterogeneity

We present additional results for attention heads and FFNs in Figures 6 to 8, confirming heterogeneity in block level activations.

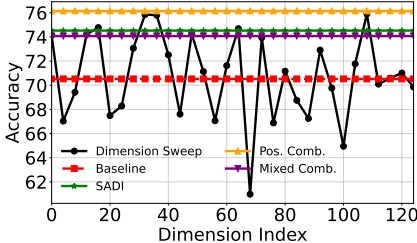

Figure 6: Steering results for 26th attention output at 15th layer. **Positive Combination**: steering four beneficial dimensions (32, 36, 48, 64). **Mixed Combination**: steering those four plus two detrimental dimensions (44, 88).

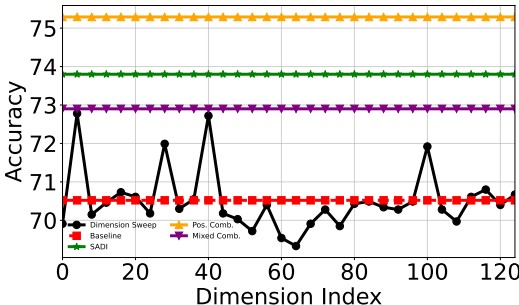

Figure 7: Steering results for 1st attention output at 19th layer. **Positive Combination**: steering three beneficial dimensions (28, 40, 100). **Mixed Combination**: steering those three plus two detrimental dimensions (64, 68).

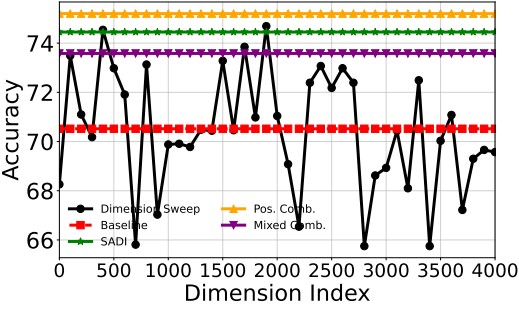

Figure 8: Steering results for the FFN output at layer 17. **Positive Combination**: steering three beneficial dimensions (400, 800, 2400). **Mixed Combination**: steering those three plus two detrimental dimensions (2200, 3500).

## B    Detailed Experiment Setup

### B.1    Contrastive Sample Construction

Contrastive sample pairs are required by AUSteer and all baseline methods. To ensure a fair comparison, we follow SADI (Wang et al., 2025b) and STA (Wang et al., 2025a) to construct same pairs

for every method. For each sample in commonsense reasoning, we form a positive sample by concatenating the question with the correct answer, and a negative sample by concatenating the question with a randomly selected incorrect answer. In math problem solving, we use the question plus the correct answer as the positive sample. For the negative sample, we use a sentence encoder to select the most semantically similar incorrect answer from the answer pool and concatenate it with the question. For detoxification, we select entries from RealToxicityPrompts with high toxicity scores as negative prompts. Following STA, the safe response is used as the positive sample. In BPO, we use the original prompt paired with a high-quality (human-preferred) response as the positive sample, and the same prompt paired with a low-quality response from the dataset as the negative sample.

We clarify that (1) contrastive samples are required by almost all activation steering methods and are a common practice in prior work (Li et al., 2023b; Rimsky et al., 2024; Wang et al., 2025b;a), rather than a limitation unique to AUSteer; (2) constructing these pairs is generally straightforward based on available samples and easy to implement; and (3) we provide and verify a simple, general, and ready-to-use procedure for constructing contrastive pairs across different and new tasks.

**(1) Contrastive samples are widely required in activation steering.** Existing activation steering methods, including ITI, CAA, SADI, and STA, all rely on contrastive positive–negative samples to localize important components and/or to estimate steering vectors. Thus, the requirement of contrastive pairs is not a limitation specific to AUSteer, but rather a standard and widely adopted practice. For fair comparison, we also ensure that all baseline methods use the same contrastive pairs in our experiments.

**(2) Constructing contrastive pairs is simple in practice.** Following prior work such as SADI and STA, constructing contrastive pairs is straightforward. For commonsense reasoning tasks, the negative sample can be obtained by pairing the question with an incorrect answer. For other datasets, negative samples can be generated by selecting semantically similar responses from a pool of candidate answers, or by using datasets that already include ready-to-use negative samples.

**(3) A general solution for new tasks.** For tasks not covered in existing studies, we use a general and effective approach. **Positive sample:** concatenate the question with the correct answer. **Negative sample:** use a sentence encoder to identify the most semantically similar *incorrect* answer from the answer pool and concatenate it with the question (Feng et al., 2025). For example, previous studies did not include math tasks, so we constructed contrastive pairs for those tasks using this method. For all other tasks, we use the contrastive pairs provided by prior work to ensure fair comparison.

**(4) Empirical verification of the general solution.** Using the above general construction method, we re-evaluated AUSteer on Llama2-7B-Chat. As shown in Table 5, this simple approach achieves performance *comparable to or even slightly better* than our original results.

Table 5: Results on LLaMA2-7B-Chat with new contrastive pairs.

| Method | Avg. Acc. (5 tasks) | Detox | BPO |
|---|---|---|---|
| Vanilla | 56.01 | – | – |
| SADI | 59.49 | 86.32 | 13.50 |
| AUSteer (previous result) | 61.34 | 89.24 | 22.00 |
| AUSteer (new solution) | 61.53 | 89.99 | 22.50 |

In summary, contrastive pairs are commonly required across activation steering studies and are not a unique limitation of AUSteer. Moreover, constructing them is straightforward, and our general solution is simple, effective, and empirically validated to yield strong performance. We acknowledge that the reliance on contrastive pairs is an inherent limitation of existing activation-steering methods, and we plan to explore approaches that reduce or eliminate this requirement in future work.

## B.2 DATA STATISTICS

Following SADI, we use at most 1,000 contrastive pairs per task to identify important MHA and FFN components or to generate steering vectors. For evaluation, we use the full test set of each task. Detailed dataset statistics are provided in Table 6.

Table 6: The number of contrastive pairs and testing samples for 7 tasks.

|  | BoolQ | COPA | WinoGrande | SVAMP | MAWPS | Detox | BPO |
|---|---|---|---|---|---|---|---|
| # of contrastive pairs | 1000 | 1000 | 1000 | 700 | 1000 | 1000 | 1000 |
| Test | 3270 | 500 | 1267 | 300 | 355 | 1199 | 200 |

### B.3 PROMPTS FOR DATASETS AND EVALUATION

To ensure a fair comparison, we use identical prompt templates across all methods. For common-sense reasoning tasks, the templates strictly follow SADI (Wang et al., 2025b) and the authors' released code. For RealToxicityPrompts, the templates follow STA (Wang et al., 2025a). Figure 9 shows the templates for SVAMP and MAWPS. For BPO, we use the prompts provided in the dataset directly.

> **SVAMP**
> Answer the following grade-school math word problem. Reply with only the final answer as a number.
> Question: {*question*}
> Answer:

> **MAWPS**
> Answer the grade school math word problem below, using step-by-step problem-solving process. Print the final answer after \"####\.
> Question: {*question*}
> Answer:

Figure 9: Prompt templates for math problems.

## C HYPERPARAMETER SENSITIVITY

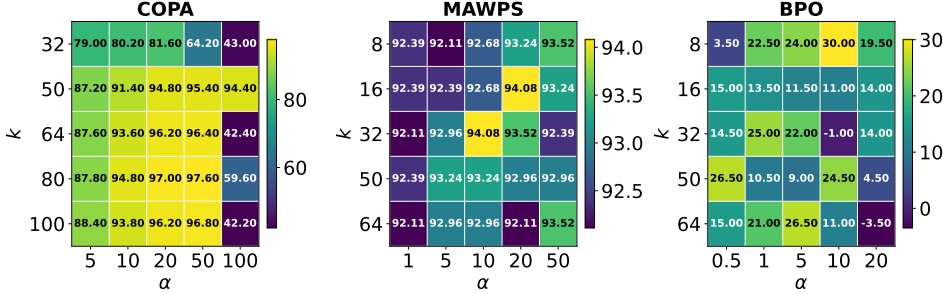

Figure 10: Performance heatmaps for COPA, MAWPS, and BPO tasks as functions of $\alpha$ and $k$.

AUSteer introduces two hyperparameters: (1) $k$, the number of AU-level activations selected for steering; and (2) $\alpha$, a global steering-strength factor. To verify the claim that we can *steer less to achieve more*, we cap the number of steered activations at 100 and sweep both $k$ and $\alpha$ from 1 to 100 for main experiments. Figure 10 reports performance across COPA, MAWPS, and BPO. Neighboring settings around the optimal hyperparameters achieve comparable results, indicating robustness. The optimal values vary across tasks to some extent, showing that the hyperparameters are task-specific, a trend consistent with Wang et al. (2025b).

To set the hyperparameters for each task, we provide two solutions: (1) under sufficient computing resources, we perform a full hyperparameter sweep, which is consistent with previous studies (Li et al., 2023b; Rimsky et al., 2024; Wang et al., 2025b;a); and (2) in computing-constrained scenarios, we recommend using a very small validation set to conduct a quick hyperparameter sweep. In addition, (3) the optimal hyperparameters used in our experiments are reported in Tables 8 and 9.

**General hyperparameter sweep (resource-sufficient case).** Task-specific hyperparameters are still a common challenge in activation steering, and the standard solution used widely in existing studies is to perform a sweep (Li et al., 2023b; Rimsky et al., 2024; Wang et al., 2025b;a). Following these studies, we perform a full hyperparameter sweep to empirically determine optimal $\alpha$ and $k$.

We also run the same sweep for all baseline methods to ensure fair comparison in Table 1. Across tasks, both $\alpha$ and $k$ typically fall within **1–100** and consistently yield strong results.

**Fast sweep using a small validation set (resource- or time-constrained case).** When resources are limited, we recommend sweeping using only **50–100 validation samples**. This process is extremely fast (e.g., $\sim$**5 minutes** on an H100 GPU for 100 samples for the COPA task). Results using this small-set search are shown in below Table 7. It can be observed that even with only very few samples for hyperparameter selection, our proposed method still significantly outperforms the baseline methods and achieves results comparable to the full search.

Table 7: Results of fast sweep on LLaMA2-7B-Chat

| Method | Avg. Acc. (5 tasks) | Detox | BPO |
|---|---|---|---|
| Vanilla | 56.01 | – | – |
| SADI | 59.49 | 86.32 | 13.50 |
| AUSteer (100-sample search) | 61.03 | 88.49 | 22.00 |
| AUSteer (Full search) | 61.34 | 89.24 | 22.00 |

The optimal values of $\alpha$ and $k$ used for each task are reported in Tables 8 and 9. These values were obtained via full sweep, and the same process was applied to baseline methods for fairness. The task-specific variation of hyperparameters aligns with observations from prior work, indicating that different tasks may require different hyperparameter values.

However, **for any given task, the hyperparameters are stable and robust**. For example, an shown in Figure 10, for the COPA task, when $20 \leq \alpha \leq 50$ and $64 \leq k \leq 100$, the performance remains stable and varies within only 1.5%, while still significantly outperforming the baseline methods. For the MAWPS task, when $10 \leq \alpha \leq 50$ and $16 \leq k \leq 50$, the performance also varies within approximately 1.5%. Therefore, for each specific task, our method is hyperparameter-robust, and within the optimal region, it achieves comparable results with only small variations.

Table 8: Optimal $\alpha$ for main experiments.

| BoolQ | COPA | Winogrande | SVAMP | MAWPS | Detoxic. | BPO |
|---|---|---|---|---|---|---|
| 15 | 50 | 100 | 8 | 8 | 15 | 32 |
| 50 | 50 | 100 | 100 | 50 | 8 | 10 |
| 10 | 20 | 20 | 10 | 50 | 10 | 16 |

Table 9: Optimal $k$ for main experiments.

| BoolQ | COPA | Winogrande | SVAMP | MAWPS | Detoxic. | BPO |
|---|---|---|---|---|---|---|
| 100 | 16 | 2 | 50 | 80 | 16 | 16 |
| 8 | 80 | 64 | 4 | 8 | 16 | 8 |
| 100 | 8 | 100 | 100 | 2 | 8 | 10 |

In summary, although the hyperparameters remain robust within an individual task, task-specific hyperparameters are still a common challenge in activation steering. The standard solution used widely in existing studies is to perform a sweep. To further reduce cost, we show that sweeping on a very small validation set is both **efficient** and **highly effective**, while still outperforming strong baselines. We will explore more principled approaches to reducing task-dependent hyperparameter sensitivity in future work.

# D CHARACTERISTICS OF ACTIVATION MOMENTUM AND LOCALIZED AUs

Figures 11a and 11b report the discriminative score $s_i$ for each AU in both MHA and FFN, computed via activation momentum. We observe pronounced heterogeneity: within attention heads and FFN blocks, some dimensions/AUs are strongly discriminative while others are not. Moreover, most AUs localize to the middle or latter layers, consistent with prior findings (Wang et al., 2025b) that middle layers support reasoning while latter layers are critical for language generation.

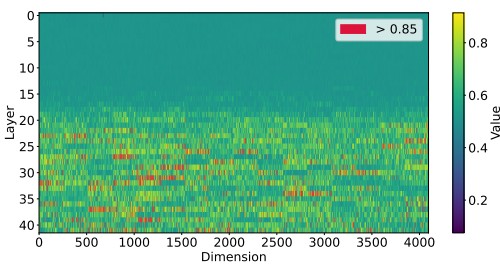 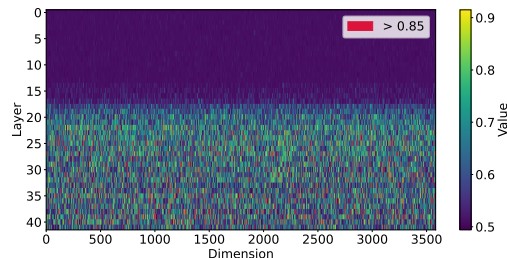

(a) AU scores in the MHA of Gemma2-9B-it on the COPA dataset.

(b) AU scores in the FFN of Gemma2-9B-it on the COPA dataset.

Figure 11: Characteristics of AUs in MHA and FFN.

# E  AU OVERLAP ACROSS TASKS

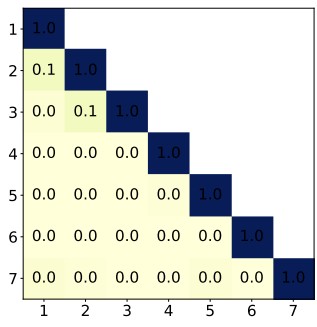 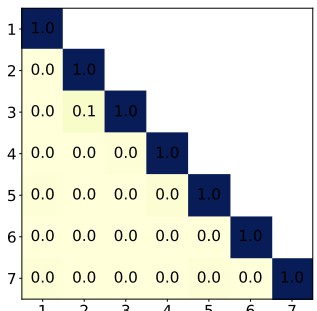

(a) Overlap of identified AUs in the MHA of Gemma2-9B-it across different tasks.

(b) Overlap of identified AUs in the FFN of Gemma2-9B-it across different tasks.

Figure 12: Overlap of localized AUs across tasks. Tasks 1–7 correspond to BoolQ, COPA, Wino-Grande, SVAMP, MAWPS, Detoxification, and BPO.

We visualize the overlap of localized AUs across tasks in Figures 12a and 12b. Only very few AUs are shared between tasks, indicating that the AUs supporting different functions are highly specialized. This pattern is consistent with prior studies (Li et al., 2023a; Wang et al., 2025b).

# F  STEERING STABILITY WITH VARYING DATA SIZE

Following prior work, AUSteer uses contrastive sample pairs to localize important AUs. We evaluate how its accuracy varies with the number of contrastive pairs. As shown in Figure 13, the accuracy on Gemma2-9B-it improves as the dataset grows. Notably, with 300–500 pairs, AUSteer achieves performance comparable to using 1,000 pairs. This demonstrates its effectiveness in low-data regimes.

# G  SCALABILITY ON MORE LLMS

To further verify the generalizability and scalability of AUSteer, we evaluate it on two representative large models with diverse structures: (1) **Qwen3-30B-A3B**, a 30B-scale **sparse MoE** model; and (2) **Llama-3.3-70B-Instruct**, where we use the **4-bit quantized** version to enable evaluation on a consumer GPU and to test AUSteer's compatibility with **heavily quantized LLMs**. The results are shown in Table 10. In most cases, AUSteer improves performance by **1%–3%**, confirming its effectiveness and scalability across larger, structurally diverse and heavily quantized LLMs.

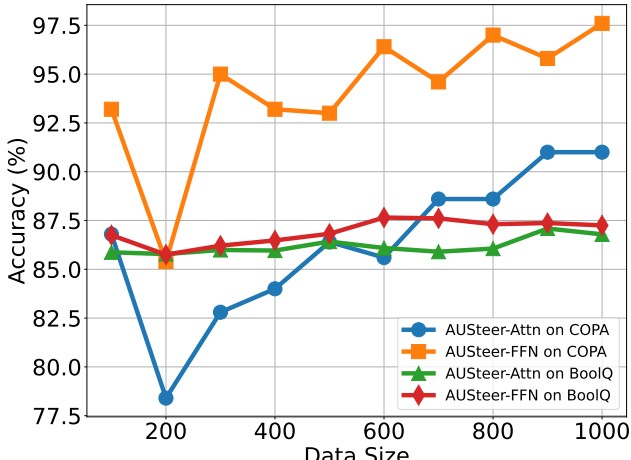

Figure 13: Relationship between accuracy and the number of contrastive pairs

Table 10: More results of diverse and larger LLMs.

| Model | Method | BoolQ | COPA | WinoG |
|-------|--------|-------|------|-------|
| **Qwen3-30B-A3B** | Vanilla | 86.82 | 93.40 | 65.98 |
|  | AUSteer | 88.69 | 97.80 | 67.17 |
| **Llama-3.3-70B-Instruct** | Vanilla | 89.54 | 98.60 | 78.14 |
|  | AUSteer | 90.67 | 99.20 | 79.95 |

## H  ANALYZING ACTIVATION MOMENTUM IN DISCRIMINATIVE CAUSALITY

We explain the connection between activation momentum and causality based on both theoretical justification and empirical evidence.

**Theoretical Justification.** Prior work in LLM interpretability (Dar et al., 2023; Geva et al., 2022; Li et al., 2024; Katz et al., 2024; Neo et al., 2025) shows that intermediate hidden states $x$ in LLMs can be directly projected to the output logits through the LM head. This projection directly affects the model's final next-token distribution. Formally, the LM head $\mathcal{M}$ computes:

$$o = \mathcal{M}x,$$

where $o$ is the vector of output logits.

This aligns with our observations in Section 3.3: different AUs govern different output token distributions, and as steering strength increases, the LLM's output tends to converge to the AU's token distribution. For a contrastive pair, the logit difference caused by the two inputs is:

$$\Delta o = o^{\text{pos}} - o^{\text{neg}}.$$

For AU $u_i$ and contrastive pair $j$, define the activation momentum:

$$m_i^j = x_i^{\text{pos}} - x_i^{\text{neg}}.$$

Based on $o = \mathcal{M}x$, we apply a first-order Taylor expansion around $x_i^{\text{neg}}$:

$$o(x_i^{\text{pos}}) \approx o(x_i^{\text{neg}}) + \frac{\partial o}{\partial x_i}\left(x_i^{\text{pos}} - x_i^{\text{neg}}\right).$$

Rearranging gives:

$$\Delta o_i^j = o^{\text{pos}} - o^{\text{neg}} \approx \frac{\partial o}{\partial x_i}m_i^j.$$

This equation shows that the change in activation of AU $u_i$ directly causes a proportional change in the output logits. Thus:

- $m_i^j > 0$ tends to increase the logit difference favoring the positive sample.

- $m_i^j < 0$ tends to favor the negative sample.

- If $m_i^j$ is **consistent across many pairs**, then the AU $u_i$ has a **stable discriminative causal effect** on the output logits.

This provides the theoretical grounding for activation momentum.

**Empirical Evidence** To further validate the effectiveness of activation momentum, we compare it against two alternatives: (i) randomly selected AUs and (ii) the activation-difference method used in SADI. On Gemma2-9B-it, the performance follows the order: 83.96 (activation momentum, ours) > 83.12 (activation difference by SADI) > 79.08 (random selection). These results demonstrate the superior performance of activation momentum. Additional experimental details are provided in Section 5.3.

To summarize, we establish the connection between activation momentum and discriminative output causality through both theoretical analysis and empirical validation, thereby grounding and verifying our method.

## I    COMPUTATION OVERHEAD ANALYSIS

We conducted a detailed efficiency and computation analysis from two perspectives: (1) smaller steering footprint, and (2) the actual computational overhead measured in practice, including activation-momentum computation time, inference-time cost, latency, and stability. Our results show that AUSteer requires **less overhead and fewer interventions** while achieving **better performance** than baseline methods.

**Smaller steering footprint**. As shown in Table 1, baseline methods typically require intervening on 3,000–4,000 activations (or $k_h \times 128$), whereas AUSteer requires at most **100** intervened activations while still achieving the best results on most tasks.

**Detailed overhead analysis**. We examine the computational overhead of our method and all baselines at each stage of the method. In the preparation phase, AUSteer extracts activations from contrastive pairs to compute activation momentum, whereas baseline methods usually require component localization and steering-vector estimation. During inference, each method applies its corresponding intervention, and we compare the resulting overhead across methods. It is worth mentioning that activation momentum calculation only requires a single forward pass over a small set of contrastive examples. No backward pass, gradient computation, model modification, or training is needed. Extracting activations simply involves reading intermediate hidden states. Therefore, for any LLM size, activation momentum can always be computed using the same GPU memory required for standard inference, since both perform identical forward passes.

Table 11 below compares the computation cost of AUSteer with ITI and SADI across six metrics: (1) GPU memory for contrastive samples of all tasks, (2) total runtime on all contrastive samples, (3) GPU memory during inference, (4) inference time over seven tasks, (5) latency, and (6) latency stability (std from five repeated trials). It is noted that all methods rely on contrastive samples to compute the necessary steering signals—whether for activation differences, localization, activation momentum (ours), or steering-vector estimation (other methods). The backbone LLM is Gemma2-9B-it (batch size = 1, GPU = NVIDIA H100).

Compared to other activation-steering baselines, AUSteer has the lowest overhead while achieving the best results. Specifically, AUSteer requires **only ∼15 minutes** to compute activation momentum and localization, **no additional GPU memory** beyond inference, and exhibits **lower overhead** than ITI and SADI while achieving **better performance**, demonstrating its computational efficiency. During inference (steering), AUSteer also requires slightly less time than the baseline methods, further demonstrating its efficiency in runtime overhead.

Table 11: Computation Overhead Comparison.

| Method | GPU Memory (contrastive) | Time (contrastive) | GPU Memory (Inference) | Inference Time | Latency | Stability |
|---|---|---|---|---|---|---|
| Vanilla LLM | – | – | 18 GB | 53 min 12 sec | 0.45 s/sample | $\Delta 0.005$ |
| ITI | 18 GB | 18 min 39 sec | 18 GB | 59 min 29 sec | 0.50 s/sample | $\Delta 0.01$ |
| SADI | 18 GB | 14 min 41 sec | 18 GB | 55 min 05 sec | 0.47 s/sample | $\Delta 0.005$ |
| AUSteer (Ours) | 18 GB | 14 min 41 sec | 18 GB | 54 min 41 sec | 0.46 s/sample | $\Delta 0.007$ |

For the **computational cost on larger LLMs such as 4-bit Llama-3.3-70B-Instruct**, taking COPA as an example, we report both the preparation (activation-momentum computation) and inference overhead. During the activation-momentum computation stage, using 1000 contrastive pairs, AUSteer requires 40 GB of GPU memory and around 15 minutes. During inference, the vanilla LLM requires 40 GB of GPU memory and 3 min 46 sec to run all test samples, while AUSteer requires 40 GB and 3 min 54 sec. These empirical results show that activation momentum scales successfully to large LLMs and remains far from computationally intensive, even on a 70B LLM.

To summarize, our proposed method requires the **least intervention footprint** and **lowest computational overhead**, while achieving the **best performance** on most tasks. This provides clear empirical evidence supporting our argument that a smaller steering footprint can achieve improved efficiency.

## J  BROADER CONTROL VARIANTS OF AUSTEER

We conducted additional experiments on broader steering variants and found that, contrary to the assumption that "steering more AUs should be better," **precise partial AU control is the correct strategy**. It offers clear advantages over steering a large portion—or all—of the AUs.

**Steering all AUs leads to consistent performance degradation**. To test whether AUSteer is merely a constrained version of a more general "steer-all-units" method, we applied AUSteer-style dynamic weights to *all* AUs (e.g., $32 \times 4096 = 131,072$ AUs in LLaMA2-7B-Chat). After extensive hyperparameter sweeps, steering all AUs still failed to outperform the vanilla model (without any steering). This matches our analysis in Section 3.3: **different AUs regulate different output distributions**, and only a small subset is task-relevant. Steering all AUs inevitably introduces strong task-irrelevant signals, effectively injecting noise into the model outputs. In contrast, partial AU steering focuses only on useful and task-relevant subspaces, yielding meaningful and targeted interventions.

**Broader AU steering does not guarantee better performance**. We further tested variants that steer increasingly large subsets of AUs. Table 12 (COPA, LLaMA2-7B-Chat) shows that steering more than 5,000 AUs results in *worse* performance than the vanilla model. This again confirms that broader steering introduces many **task-irrelevant or harmful output distributions**, degrading performance. These findings also align with our results in Section 3.2, where steering certain AUs leads to negative effects.

Table 12: Experimental results on steering broader AUs using COPA and LLaMA2-7B-Chat. There are $32 \times 4096 = 131,072$ AUs in total in LLaMA2-7B-Chat.

| # of AUs | 0 (vanilla) | < 100 | 200 | 500 | 1000 | 3000 | 5000 | 10000 |
|---|---|---|---|---|---|---|---|---|
| Accuracy (%) | 70.8 | 82.8 | 77.2 | 73.2 | 70.8 | 70.6 | 70.4 | 70.4 |

Overall, our experiments demonstrate that **AUSteer should only steer task-relevant or beneficial AUs**, rather than steering a broad or full set of units. Partial AU control is therefore **not** a restricted version of a more general steering method—it is the **correct and uniquely effective** strategy for activation steering in LLMs.

## K  PROMOTION VERSUS SUPPRESSION

To determine whether we should promote useful AUs or suppress unhelpful ones, we conduct both empirical and theoretical analyses and show that promotion consistently outperforms suppression.

**Empirical evidence**. To evaluate the "suppression" strategy, we use AU importance scores to identify the least important AUs and apply a decreasing factor to suppress their activations. We vary the number of suppressed AUs from 0% to 99.95%, search decreasing factors from 0.05 to 0.99, and report the best results in Table 13. Experiments are conducted on LLaMA2-7B-Chat using three commonsense reasoning datasets. The results show that although suppression can yield improvements over the vanilla model, it consistently underperforms compared to the promotion-based steering used in AUSteer.

Table 13: Experimental results of suppressing AUs.

| Method | BoolQ | COPA | WinoG. |
|---|---|---|---|
| Vanilla | 70.52 | 70.8 | 50.91 |
| Suppression | 73.36 | 71.6 | 53.12 |
| Promotion (AUSteer, ours) | 75.57 | 82.8 | 53.28 |

**Theoretical explanation**. Prior work (Geva et al., 2022; Dar et al., 2023) shows that LLMs update predictions primarily through a **promotion mechanism**, where top-candidate tokens are driven by dominant positive sub-updates rather than by suppressing irrelevant ones. Consequently, directly *promoting* task-relevant AUs aligns better with the model's intrinsic update dynamics, producing stronger and more targeted effects than suppression.

