# OpenReview forum: "Fine-Grained Activation Steering: Steering Less, Achieving More"
_ICLR.cc/2026/Conference — ICLR 2026 Poster_

### Official Review · Reviewer_xSGu · 2025-11-01

**Soundness:** 3
**Presentation:** 3
**Contribution:** 2
**Rating:** 4
**Confidence:** 4

**Summary:**

The paper proposes AUSteer, a method for fine-grained activation steering in large language models. Instead of steering at the block level, AUSteer operates at the Atomic Unit (AU) level, corresponding to individual activation dimensions. The authors show that block-level activations mix helpful and harmful components, making coarse interventions inefficient. AUSteer identifies discriminative AUs through an activation momentum metric computed from contrastive pairs and applies adaptive per-AU scaling. Experiments on seven benchmarks and three model families demonstrate consistent improvements over block-level steering with far fewer activations, suggesting that steering less can achieve more.

**Strengths:**

The problem is clearly defined and relevant. The idea of decomposing block activations into AUs is intuitive and well motivated. AUSteer is simple, interpretable, and does not require retraining. The experiments are broad and consistent across tasks and models, and the analysis convincingly shows heterogeneity within block activations.

**Weaknesses:**

1) Efficiency claim lacks evidence:

The paper’s argument that a smaller steering footprint improves efficiency is not empirically verified. No inference-time or computational measurements are provided, and efficiency is used only in a representational sense.


2) Lack of comparison with broader control variants.

The paper assumes that steering only a subset of AUs is inherently superior, but does not test a broader or fully generalized steering scheme where all AUs are jointly optimized or selectively weighted. Without such a comparison, it remains unclear whether partial AU control offers unique advantages beyond being a constrained version of more general steering

**Questions:**

1) Does efficiency refer to computational speed or representational precision?
2) Have you measured inference cost, latency, or stability?
3) Would steering all AUs with selective suppression perform similarly?
4) How scalable is activation momentum computation for very large models?

---

> ### Author Response · Authors · 2025-11-19
> **Response 1/N**
>
> Dear Reviewer xSGu,
>
> Thank you for taking the time to review our paper and for recognizing its strengths, including the **intuitive and well-motivated fine-grained decomposition, the simplicity and interpretability of the proposed method, and the comprehensive experiments**. Your comments mainly focused on clarifying efficiency and overhead justification, expanding comparisons with broader control variants of AUSteer, and discussing scalability. We have carefully addressed all of your concerns and incorporated the corresponding discussions and clarifications into the revised paper. We are pleased to see that, with your insightful comments and suggestions, the quality of the paper has been significantly improved. Our detailed responses [**R**] to your raised weaknesses [**W**] and questions [**Q**] are provided below.
>
> [**W1**] Efficiency claim lacks evidence. The paper’s argument that a smaller steering footprint improves efficiency is not empirically verified. No inference-time or computational measurements are provided, and efficiency is used only in a representational sense.
>
> [**R1**] Thank you for pointing this out. We conducted a detailed efficiency analysis from two perspectives: (1) smaller steering footprint, and (2) the actual computational overhead measured in practice, including activation-momentum computation time, inference-time cost, latency, and stability. Our results show that AUSteer requires **less overhead and fewer interventions** while achieving **better performance** than baseline methods. These analyses have been added to Appendix I in the revised paper. The detailed results are summarized below.
>
> (1) **Smaller steering footprint.** As shown in Table 1 of our paper, baseline methods typically require intervening on 3,000–4,000 activations (or $k_h \times 128$), whereas AUSteer requires at most **100** intervened activations while still achieving the best results on most tasks.
>
>
> (2) **Detailed overhead analysis.** We examine the computational overhead of our method and all baselines at each stage of the method. In the preparation phase, AUSteer extracts activations from contrastive pairs to compute activation momentum, whereas baseline methods usually require component localization and steering-vector estimation. During inference, each method applies its corresponding intervention, and we compare the resulting overhead across methods.
>
>
> Table R1 below compares the computation cost of AUSteer with ITI and SADI across six metrics: 1) GPU memory for contrastive samples of all tasks, 2) total runtime on all contrastive samples, 3) GPU memory during inference, 4) inference time over seven tasks, 5) latency, and 6) latency stability (std from five repeated trials). It is noted that all methods rely on contrastive samples to compute the necessary steering signals—whether for activation differences, localization, activation momentum (ours), or steering-vector estimation (other methods).
> The backbone LLM is Gemma2-9B-it (batch size = 1, GPU = NVIDIA H100).
>
> Compared to other activation-steering baselines, AUSteer has the lowest overhead while achieving the best results. Specifically, AUSteer requires **only ~15 minutes** to compute activation momentum and localization, **no additional GPU memory** beyond inference, and exhibits **lower overhead** than ITI and SADI while achieving **better performance**, demonstrating its computational efficiency. During inference (steering), AUSteer also requires slightly less time than the baseline methods, further demonstrating its efficiency in runtime overhead.
>
> **Table R1: Overhead Comparison**
>
> | Method           | GPU Memory (on contrastive samples) | Required Time (on contrastive samples) | GPU Memory (Inference) | Inference Time | Latency        | Stability     |
> |------------------|---------------------------|------------------------------|--------------------------|----------------|-----------------|----------------|
> | Vanilla LLM      | -                      | –                            |  18 GB               | 53 min 12 sec  | 0.45 s/sample  | Δ 0.005        |
> | ITI              | 18 GB                     | 18 min 39 sec                | 18 GB                   | 59 min 29 sec  | 0.50 s/sample  | Δ 0.01         |
> | SADI             | 18 GB                     | 14 min 41 sec                | 18 GB                   | 55 min 05 sec  | 0.47 s/sample  | Δ 0.005        |
> | AUSteer (Ours)   | 18 GB                     | 14 min 41 sec                | 18 GB                   | 54 min 41 sec  | 0.46 s/sample  | Δ 0.007        |
>
> To summarize, our proposed method requires the **least intervention footprint** and **lowest computational overhead**, while achieving the **best performance** on most tasks. This provides clear empirical evidence supporting our argument that a smaller steering footprint can achieve improved efficiency.

---

> ### Author Response · Authors · 2025-11-19
> **Response 2/N**
>
> [**W2**] Lack of comparison with broader control variants. The paper assumes that steering only a subset of AUs is inherently superior, but does not test a broader or fully generalized steering scheme where all AUs are jointly optimized or selectively weighted. Without such a comparison, it remains unclear whether partial AU control offers unique advantages beyond being a constrained version of more general steering.
>
> [**R2**] Thank you for raising this concern. This question allows us to further analyze and clarify the internal mechanism of AUSteer. To address it, we conducted additional experiments on broader steering variants and found that, contrary to the assumption that “steering more AUs should be better,” **precise partial AU control is the correct strategy**. It offers clear advantages over steering a large portion—or all—of the AUs. Our findings are summarized below, and the full analysis has been added to Appendix J.
>
> (1) **Steering all AUs leads to consistent performance degradation.** To test whether AUSteer is merely a constrained version of a more general “steer-all-units” method, we applied AUSteer-style dynamic weights to *all* AUs (e.g., 32 × 4096 = 131,072 AUs in LLaMA2-7B-Chat). After extensive hyperparameter sweeps, steering all AUs still failed to outperform the vanilla model (without any steering). This matches our analysis in Section 3.3: **different AUs regulate different output distributions, and only a small subset is task-relevant.** Steering all AUs inevitably introduces strong task-irrelevant signals, injecting noise into the model outputs. In contrast, partial AU steering focuses only on useful and task-relevant subspaces, yielding meaningful and targeted interventions.
>
> (2) **Broader AU steering does not guarantee better performance.** We further tested variants that steer increasingly large subsets of AUs. Table R2 (COPA, LLaMA2-7B-Chat) shows that steering more than 5,000 AUs results in *worse* performance than the vanilla model. This again confirms that broader steering introduces many **task-irrelevant or harmful output distributions**, degrading performance. These findings also align with our results in Section 3.2, where steering certain AUs leads to negative effects.
>
> **Table R2** Experimental results on steering broader AUs using COPA and LLaMA2-7B-chat. There are 32 × 4096 = 131,072 AUs in total in LLaMA2-7B-Chat.
>
> | # of AUs        | 0 (vanilla) |$<$ 100 | 200  | 500  | 1000 | 3000 | 5000 | 10000 |
> |-----------------|-------------|------|------|------|------|------|------|-------|
> | Accuracy (%)    | 70.8        | 82.8 | 77.2 | 73.2 | 70.8 | 70.6 | 70.4 | 70.4  |
>
>
> Overall, our experiments demonstrate that **AUSteer should only steer task-relevant or beneficial AUs**, rather than steering a broad or full set of units. Partial AU control is therefore **not** a restricted version of a more general steering method—it is the **correct and uniquely effective** strategy for activation steering in LLMs.

---

> > ### Author Response · Authors · 2025-11-19
> > **Response 3/N**
> >
> > [**Q1 & Q2**] Does efficiency refer to computational speed or representational precision? Have you measured inference cost, latency, or stability?
> >
> > [**R3**] Thank you for your questions. We apologize for not clearly defining efficiency in the original submission. In our work, **efficiency refers to both the reduced intervention footprint and the lower computational overhead**, while improving model performance. We have directly measured and compared inference cost, latency, and stability across methods. These results were provided in our earlier response [**R1**], and for convenience, we restate the key discussions below.
> >
> > We conducted a detailed efficiency analysis from two perspectives: (1) smaller steering footprint, and (2) the actual computational overhead measured in practice, including activation-momentum computation time, inference-time cost, latency, and stability. Our results show that AUSteer requires **less overhead and fewer interventions** while achieving **better performance** than baseline methods. These analyses have been added to Appendix G in the revised paper. The detailed results are summarized below.
> >
> > (1) **Smaller steering footprint.** As shown in Table 1 of our paper, baseline methods typically require intervening on 3,000–4,000 activations (or $k_h \times 128$), whereas AUSteer requires at most **100** intervened activations while still achieving the best results on most tasks.
> >
> >
> > (2) **Detailed overhead analysis.** We examine the computational overhead of our method and all baselines at each stage of the method. In the preparation phase, AUSteer extracts activations from contrastive pairs to compute activation momentum, whereas baseline methods usually require component localization and steering-vector estimation. During inference, each method applies its corresponding intervention, and we compare the resulting overhead across methods.
> >
> >
> > Table R3 below compares the computation cost of AUSteer with ITI and SADI across six metrics: 1) GPU memory for contrastive samples of all tasks, 2) total runtime on all contrastive samples, 3) GPU memory during inference, 4) inference time over seven tasks, 5) latency, and 6) latency stability (std from five repeated trials). It is noted that all methods rely on contrastive samples to compute the necessary steering signals—whether for activation differences, localization, activation momentum (ours), or steering-vector estimation (other methods).
> > The backbone LLM is Gemma2-9B-it (batch size = 1, GPU = NVIDIA H100).
> >
> > Compared to other activation-steering baselines, AUSteer has the lowest overhead while achieving the best results. Specifically, AUSteer requires **only ~15 minutes** to compute activation momentum and localization, **no additional GPU memory** beyond inference, and exhibits **lower overhead** than ITI and SADI while achieving **better performance**, demonstrating its computational efficiency. During inference (steering), AUSteer also requires slightly less time than the baseline methods, further demonstrating its efficiency in runtime overhead.
> >
> > **Table R3: Overhead Comparison**
> >
> > | Method           | GPU Memory (on contrastive samples) | Required Time (on contrastive samples) | GPU Memory (Inference) | Inference Time | Latency        | Stability     |
> > |------------------|---------------------------|------------------------------|--------------------------|----------------|-----------------|----------------|
> > | Vanilla LLM      | -                      | –                            |  18 GB               | 53 min 12 sec  | 0.45 s/sample  | Δ 0.005        |
> > | ITI              | 18 GB                     | 18 min 39 sec                | 18 GB                   | 59 min 29 sec  | 0.50 s/sample  | Δ 0.01         |
> > | SADI             | 18 GB                     | 14 min 41 sec                | 18 GB                   | 55 min 05 sec  | 0.47 s/sample  | Δ 0.005        |
> > | AUSteer (Ours)   | 18 GB                     | 14 min 41 sec                | 18 GB                   | 54 min 41 sec  | 0.46 s/sample  | Δ 0.007        |
> >
> > To summarize, our proposed method requires the **least intervention footprint** and **lowest computational overhead**, while achieving the **best performance** on most tasks. This provides clear empirical evidence supporting our argument that a smaller steering footprint can achieve improved efficiency.

---

> ### Author Response · Authors · 2025-11-19
> **Response N/N**
>
> [**Q3**] Would steering all AUs with selective suppression perform similarly?
>
> [**R4**] Thank you for your question. We conducted additional experiments and found that steering AUs with selective suppression performs **slightly worse** than promoting beneficial AUs, as done in AUSteer. We provide both empirical and theoretical justification below, and these discussions have been added to Appendix K in the revised paper.
>
> (1) **Empirical evidence.** To evaluate the “suppression” strategy, we use AU importance scores to identify the least important AUs and apply a decreasing factor to suppress their activations. We vary the number of suppressed AUs from 0% to 99.95%, search decreasing factors from 0.05 to 0.99, and report the best results in Table R4. Experiments are conducted on LLaMA2-7B-Chat using three commonsense reasoning datasets. The results show that although suppression can yield improvements over the vanilla model, it consistently underperforms compared to the promotion-based steering used in AUSteer.
>
> **Table R4** Experimental results of suppressing AUs.
>
> | Method       | BoolQ | COPA | WinoG. |
> |--------------|-------|------|------|
> | Vanilla      | 70.52 | 70.8 | 50.91 |
> | Suppression  | 73.36 | 71.6 | 53.12 |
> | Promotion (AUSteer, ours)    | 75.57 | 82.8 | 53.28 |
>
>
> (2) **Theoretical explanation.** Prior work [1,2] shows that LLMs update predictions primarily through a **promotion mechanism**, where top-candidate tokens are driven by dominant positive sub-updates rather than by suppressing irrelevant ones. Consequently, directly *promoting* task-relevant AUs aligns better with the model’s intrinsic update dynamics, producing stronger and more targeted effects than suppression.
>
> References
>
> [1] Transformer Feed-Forward Layers Build Predictions by Promoting Concepts in the Vocabulary Space, EMNLP 2023
>
> [2] Analyzing transformers in embedding space, ACL 2023
>
> [**Q4**] How scalable is activation momentum computation for very large models?
>
> [**R5**] Thank you for your question. We would like to clarify that the computation of **activation momentum is highly efficient in practice**, **not resource-intensive**, and can be **successfully scaled to larger LLMs**. The detailed analysis is provided below.
>
>
> (1) Firstly, activation momentum does not require extra GPU memory. Activation momentum only requires a **single forward pass** over a small set of contrastive examples. No backward pass, gradient computation, model modification, or training is needed. Extracting activations simply involves reading intermediate hidden states. Therefore, for any LLM size, activation momentum can always be computed using the **same GPU memory** required for standard inference, since both perform identical forward passes.
>
> (2) Furthermore, we report the performance and the computation overhead on lager LLMs. We evaluate it on two representative large models with diverse structures: (i) **Qwen3-30B-A3B**, a 30B-scale **sparse MoE** model; and (ii) **Llama-3.3-70B-Instruct**, where we use the **4-bit quantized** version to enable evaluation on a consumer GPU and to test AUSteer’s compatibility with **heavily quantized LLMs**. The results are shown in Table R5 below. In most cases, AUSteer improves performance by **1%–3%**, confirming its effectiveness and scalability across larger, structurally diverse and heavily quantized LLMs.
>
> (3) For the **computational cost on Llama-3.3-70B-Instruct**, taking COPA as an example, we report both the preparation (activation-momentum computation) and inference overhead. During the activation-momentum computation stage, using 1000 contrastive pairs, AUSteer requires 40 GB of GPU memory and around 15 minutes. During inference, the vanilla LLM requires 40 GB of GPU memory and 3 min 46 sec to run all test samples, while AUSteer requires 40 GB and 3 min 54 sec. These empirical results show that activation momentum **scales successfully to large LLMs**.
>
>
> **Table R5: More results of diverse and larger LLMs.**
>
> | Model   | Method   | BoolQ | COPA | WinoG |
> |---|---|--|----|--|
> | **Qwen3-30B-A3B**  | Vanilla  | 86.82 | 93.4 | 65.98 |
> |     | AUSteer  | 88.69 | 97.8 | 67.17 |
> | **Llama-3.3-70B-Instruct** | Vanilla  | 89.54 | 98.6 | 78.14 |
> |     | AUSteer  | 90.67 | 99.2 | 79.95 |
>
> **In summary**, activation momentum is **computationally efficient**, **requires no extra GPU memory**, and **scales effectively to large models such as LLaMA-3.3-70B**.
>
> ---
>
> **Thank you again for taking the time to review our paper. All of the discussed clarifications and additional analyses have been incorporated into the revised version. We sincerely appreciate your insightful feedback and look forward to any further comments you may have.**

---

### Official Review · Reviewer_mmMY · 2025-11-02

**Soundness:** 3
**Presentation:** 3
**Contribution:** 2
**Rating:** 4
**Confidence:** 4

**Summary:**

This paper introduces AUSteer, a novel fine-grained activation steering method for LLMs that operates at the atomic unit level rather than the traditional block level (e.g., attention, FFN, or residual blocks). The authors identify a key limitation in existing steering methods: block-level activations are heterogeneous, mixing beneficial, irrelevant, and harmful components. As a result, conventional approaches (like CAA, SADI, or ITI) that steer all dimensions of a block simultaneously are coarse, inefficient, and potentially harming model performance.

To address this, AUSteer decomposes each block into fine-grained AU-level activations, where each AU corresponds to a single column of the weight matrix and each activation is a scalar. The method consists of two main components:

-AU Localization via Activation Momentum: A metric that measures the discriminative power of each AU across positive and negative contrastive samples. It identifies which AUs consistently promote or suppress desirable activations.
-Adaptive Steering : Instead of applying a fixed vector, AUSteer adjusts steering strength per input and per AU, scaling the intervention by the activation’s current value and discriminative score.

Experiments are conducted on various LLMs (LLaMA2, Gemma2, Qwen3) and tasks, including commonsense reasoning, math problem-solving, and open-ended generation.

**Strengths:**

- Clearly identifies a fundamental issue: heterogeneity in block activations—and systematically decomposes it into atomic units.

- Introduces the concept of activation momentum to measure discriminative importance without training.

- Extensive experiments across three model families (LLaMA, Gemma, Qwen) and multiple tasks (reasoning, math, safety, alignment).

- No retraining or fine-tuning required.

- Ablation studies isolate the contribution of both components.

**Weaknesses:**

- The formal derivation connecting activation momentum to discriminative causality is unclear.

- AUSteer requires carefully curated positive–negative pairs, which may not be available or trivial to construct for all tasks.

- While steering itself is efficient, computing activation momentum across many AUs and samples may still be computationally intensive for very large models.

- Hyperparameter sensitivity is unclear and needs further demonstrations and explanations.

- One wonders what is the runtime overhead for AU localization and steering per sample compared to block-level methods like SADI?

**Questions:**

Please see the weaknesses.

---

> ### Author Response · Authors · 2025-11-19
> **Response 1/N**
>
> Dear Reviewer mmMY,
>
> Thank you for taking the time to review our paper and for recognizing the **novelty** of our method, including our **identification of activation heterogeneity, the decomposition into atomic units, and the extensive experiments** supporting our approach. Your concerns mainly focus on the derived connection between activation momentum and causality, contrastive pair construction, computational efficiency, hyperparameter analysis, and runtime overhead. We have carefully addressed all of your concerns and incorporated the corresponding revisions into the updated paper. We are pleased that, with your suggestions, the overall quality of the paper has been greatly improved. Our responses **[R]** to the mentioned weaknesses **[W]** are detailed as follows.
>
> [**W1**] The formal derivation connecting activation momentum to discriminative causality is unclear.
>
> [**R1**] Thank you for your question. We explain the connection between activation momentum and causality based on both theoretical justification and empirical evidence. These clarification have also been added to Appendix H of our revised paper.
>
> (1) **Theoretical Justification: Connecting Activation Momentum to Output Causality**
>
> Prior work in LLM interpretability [1,2,3,4,5] and logit-lens [6] analysis shows that intermediate hidden states $x$ in LLMs can be directly projected to the output logits through the LM head. This projection directly affects the model’s final next-token distribution. Formally, the LM head $\mathcal{M}$ computes: $o = \mathcal{M}x$, where \(o\) is the vector of output logits. This aligns with our observations in Section 3.3: different AUs govern different output token distributions, and as steering strength increases, the LLM’s output tends to converge to the AU’s token distribution. For a contrastive pair, the logit difference caused by the two inputs is: $\Delta o = o^{\text{pos}} - o^{\text{neg}} $.
>
> For AU $u_i$ and contrastive pair $j$, define the activation momentum: $m_i^j = x_i^{\text{pos}} - x_i^{\text{neg}}$. Based on  $o = \mathcal{M}x$, we apply a first-order Taylor expansion around $x_i^{\text{neg}}$:
>
> $$
> o(x_i^{\text{pos}}) \approx o(x_i^{\text{neg}}) + \frac{\partial o}{\partial x_i}\left( x_i^{\text{pos}} - x_i^{\text{neg}} \right).
> $$
>
> Rearranging gives: $\Delta o_i^j = o^{\text{pos}} - o^{\text{neg}}\approx\frac{\partial o}{\partial x_i} m_i^j .$
>
> This equation shows that the change in activation momentum directly causes a proportional change in the output logits.
> Thus:
>
> - $m_i^j > 0$ tends to increase the logit difference favoring the positive sample.
> - $m_i^j < 0$ tends to favor the negative sample.
> - If $m_i^j$ is **consistent across many pairs**, then the AU $u_i$ has a **stable discriminative causal effect** on the output logits.
>
> This provides the theoretical grounding for activation momentum.
>
> (2) **Empirical Evidence.** To further validate the effectiveness of activation momentum, we compare it against two alternatives: (i) randomly selected AUs and (ii) the activation-difference method used in SADI [7]. On Gemma2-9B-it, the performance follows the order: 83.96 (activation momentum, ours) > 83.12 (activation difference by SADI) > 79.08 (random selection). These results demonstrate the superior performance of activation momentum. Additional experimental details are provided in Section 5.3 of our paper.
>
> To summarize, we establish the connection between activation momentum and discriminative output causality through both theoretical analysis and empirical validation, thereby grounding and verifying our method.
>
>
> References
>
> [1] Analyzing transformers in embedding space, ACL 2023
>
> [2] Transformer Feed-Forward Layers Build Predictions by Promoting Concepts in the Vocabulary Space, EMNLP 2022
>
> [3] Understanding and Patching Compositional Reasoning in LLMs, ACL 2024
>
> [4] Backward Lens: Projecting Language Model Gradients into the Vocabulary Space, EMNLP 2024
>
> [5] TOWARDS INTERPRETING VISUAL INFORMATION PROCESSING IN VISION-LANGUAGE MODELS, ICLR 2025
>
> [6] Interpreting gpt: the logit lens, 2020
>
> [7] Wang, Weixuan, et al. Semantics-adaptive activation intervention for llms via dynamic steering vectors. ICLR 2025.

---

> > ### Author Response · Authors · 2025-11-19
> > **Response 2/N**
> >
> > [**W2**] AUSteer requires carefully curated positive–negative pairs, which may not be available or trivial to construct for all tasks.
> >
> > [**A2**] Thank you for raising this concern. We would like to clarify that (1) contrastive samples are required by almost all activation steering methods and are a common practice in prior work [8-11], rather than a limitation unique to AUSteer; (2) constructing these pairs is generally straightforward based on available samples and easy to implement; and (3) we provide and verify a simple, general, and ready-to-use procedure for constructing contrastive pairs across different and new tasks. We elaborate on these points below. These clarifications have been added to Appendix B.1 in the revised paper.
> >
> > **(1) Contrastive samples are widely required in activation steering.** Existing activation steering methods, including ITI [8], CAA [9], SADI [10], and STA [11], all rely on contrastive positive–negative samples to localize important components and/or to estimate steering vectors. Thus, the requirement of contrastive pairs is not a limitation specific to AUSteer, but rather a standard and widely adopted practice. For fair comparison, we also ensure that all baseline methods use the same contrastive pairs in our experiments.
> >
> > **(2) Constructing contrastive pairs is simple in practice.** Following prior work such as SADI [10] and STA [11], constructing contrastive pairs is straightforward. For commonsense reasoning tasks, the negative sample can be obtained by pairing the question with an incorrect answer. For other datasets, negative samples can be generated by selecting semantically similar responses from a pool of candidate answers, or by using datasets that already include ready-to-use negative samples.
> > This process is easy to implement, and additional details are provided in Appendix B.1.
> >
> > **(3) A general solution for new tasks.** For tasks not covered in existing studies, we propose a general and effective approach. **Positive sample:** concatenate the question with the correct answer. **Negative sample:** use a sentence encoder to identify the most semantically similar *incorrect* answer from the answer pool and concatenate it with the question.
> > For example, previous studies did not include math tasks, so we constructed contrastive pairs for those tasks using this method. For all other tasks, we use the contrastive pairs provided by prior work to ensure fair comparison.
> >
> > **(4) Empirical verification of the general solution.** Using the above general construction method, we re-evaluated AUSteer on Llama2-7B-Chat. As shown in Table R1, this simple approach achieves performance *comparable to or even slightly better* than our original results.
> >
> > **Table R1. Results on Llama2-7B-Chat with new contrastive pairs**
> >
> > | Method                                   | Avg. Acc. (5 tasks) | Detox | BPO  |
> > |------------------------------------------|----------------------|--------|------|
> > | Vanilla                                  | 56.01               | –      | –    |
> > | SADI                                     | 59.49               | 86.32 | 13.50 |
> > | AUSteer (previous result)                | 61.34               | 89.24 | 22.00 |
> > | AUSteer (new solution)                   | 61.53               | 89.99 | 22.50 |
> >
> > In summary, contrastive pairs are commonly required across activation steering studies and are not a unique limitation of AUSteer. Moreover, constructing them is straightforward, and our general solution is simple, effective, and empirically validated to yield strong performance. We acknowledge that the reliance on contrastive pairs is an inherent limitation of existing activation-steering methods, and we plan to explore approaches that reduce or eliminate this requirement in future work.
> >
> > **References**
> > [8] Li, Kenneth, et al. *Inference-time intervention: Eliciting truthful answers from a language model.* NeurIPS 2023.
> >
> > [9] Rimsky, Nina, et al. *Steering Llama 2 via contrastive activation addition.* ACL2024.
> >
> > [10] Wang, Weixuan, et al. *Semantics-adaptive activation intervention for llms via dynamic steering vectors.* ICLR 2025.
> >
> > [11] Wang, Mengru, etal. *Beyond Prompt Engineering: Robust Behavior Control in LLMs via Steering Target Atoms*. ACL 2025.

---

> ### Author Response · Authors · 2025-11-19
> **Response 3/N**
>
> [**W3**] While steering itself is efficient, computing activation momentum across many AUs and samples may still be computationally intensive for very large models.
>
> [**A3**] Thank you for raising this concern. We would like to clarify that the computation of activation momentum is **highly efficient in practice** and **not resource-intensive**, even for very large models. We address this point from four perspectives: required GPU memory, runtime on contrastive samples, experimental comparison to other methods, and new results and computation overhead on LLaMA-3.3-70B. The overhead analysis has been added to Appendix I of our revised paper.
>
> **(1) Activation momentum does not require extra GPU memory.** Activation momentum only requires a **single forward pass** over a small set of contrastive examples. No backward pass, gradient computation, model modification, or training is needed. Extracting activations simply involves reading intermediate hidden states. Therefore, for any LLM size, activation momentum can always be computed using the **same GPU memory** required for standard inference, since both perform identical forward passes.
>
> **(2) Runtime on contrastive samples is flexible and scalable.**  As shown in Appendix F, AUSteer achieves comparable performance even with only a few hundred contrastive samples, instead of thousands. This allows users to flexibly control the computation cost.
>
> **(3) Compared to other activation-steering baselines, AUSteer has the lowest overhead while achieving the best results.** We examine the computational overhead of our method and all baselines at each stage of the method. In the preparation phase, AUSteer extracts activations from contrastive pairs to **compute activation momentum**, whereas baseline methods usually require **component localization and steering-vector estimation**. During inference, each method applies its corresponding intervention, and we compare the resulting overhead across methods.
>
> Table R2 compares the computation cost of AUSteer with ITI and SADI across six metrics: 1) GPU memory for contrastive samples of all tasks, 2) total runtime on all contrastive samples, 3) GPU memory during inference, 4) inference time over seven tasks, 5) latency, and 6) latency stability (std from five repeated trials). The backbone LLM is Gemma2-9B-it (batch size = 1, GPU = NVIDIA H100).
>
> AUSteer requires **only ~15 minutes** to compute activation momentum, **no additional GPU memory** beyond inference, and exhibits **lower overhead** than ITI and SADI while achieving **better performance**, demonstrating its computational efficiency. During inference (steering), AUSteer also requires slightly less time than the baseline methods, further demonstrating its efficiency in runtime overhead.
>
> **Table R2: Overhead Comparison**
>
> | Method    | GPU Memory (on contrastive samples) | Required Time (on contrastive samples) | GPU Memory (Inference) | Inference Time | Latency  | Stability   |
> |---------|----------|-------|-----|-------|------|------|
> | Vanilla LLM      | -     | –    |  18 GB   | 53 min 12 sec  | 0.45 s/sample  | Δ 0.005  |
> | ITI      | 18 GB  | 18 min 39 sec | 18 GB    | 59 min 29 sec  | 0.50 s/sample  | Δ 0.01  |
> | SADI     | 18 GB    | 14 min 41 sec  | 18 GB    | 55 min 05 sec  | 0.47 s/sample  | Δ 0.005    |
> | AUSteer (Ours)   | 18 GB  | 14 min 41 sec | 18 GB      | 54 min 41 sec  | 0.46 s/sample  | Δ 0.007   |
>
> **(4) Results and overhead on large LLMs (70B).** we evaluate it on two representative large models with diverse structures: (1) **Qwen3-30B-A3B**, a 30B-scale **sparse MoE** model; and (2) **Llama-3.3-70B-Instruct** (4-bit quantized). The results in Table R3 confirms its effectiveness and scalability across larger LLMs.
>
> For the **computational cost on 4-bit Llama-3.3-70B-Instruct**, taking COPA as an example, we report both the preparation (activation-momentum computation) and inference overhead. During the activation-momentum computation stage, using 1000 contrastive pairs, AUSteer requires 40 GB of GPU memory and around 15 minutes. During inference, the vanilla LLM requires 40 GB of GPU memory and 3 min 46 sec to run all test samples, while AUSteer requires 40 GB and 3 min 54 sec. These empirical results show that activation momentum **scales successfully to large LLMs** and remains **far from computationally intensive**, even on a 70B LLM.
>
> **Table R3: More results of diverse and larger LLMs.**
>
> | Model   | Method   | BoolQ | COPA | WinoG |
> |---|---|--|----|--|
> | **Qwen3-30B-A3B**  | Vanilla  | 86.82 | 93.4 | 65.98 |
> |     | AUSteer  | 88.69 | 97.8 | 67.17 |
> | **Llama-3.3-70B-Instruct** | Vanilla  | 89.54 | 98.6 | 78.14 |
> |     | AUSteer  | 90.67 | 99.2 | 79.95 |
>
> **In summary**, activation momentum is **computationally efficient**, **requires no extra GPU memory**, **runs quickly on contrastive samples**, **outperforms prior methods with lower overhead**, and **scales effectively to large models such as LLaMA-3.3-70B**.

---

> ### Author Response · Authors · 2025-11-19
> **Response 4/N**
>
> [**W4**] Hyperparameter sensitivity is unclear and needs further demonstrations and explanations.
>
> [**A4**] Thank you for pointing this out. In our paper, we have visualized and discussed the hyperparameter sensitivity across different tasks in Appendix C. We find that while the optimal hyperparameters are **task-specific**, they remain **stable and robust within each individual task**. To further address your comment, we provide additional discussion here, and these clarifications have been added to Appendix C in the revised version.
>
> To set the hyperparameters for each task, we provide two solutions: (1) under sufficient computing resources, we perform a full hyperparameter sweep, which is consistent with previous studies [8,9,10,11]; and (2) in computing-constrained scenarios, we recommend using a very small validation set to conduct a quick hyperparameter sweep. In addition, (3) we also analyze the hyperparameter sensitivity of different tasks.
>
> (1) **General hyperparameter sweep (resource-sufficient case). Task-specific hyperparameters are still a common challenge in activation steering, and the standard solution used widely in existing studies is to perform a sweep [8-11].** Following [8-11], we perform a full hyperparameter sweep to empirically determine optimal $\alpha$ and $k$. We also run the same sweep for all baseline methods to ensure fair comparison in Table 1. Across tasks, both $\alpha$ and $k$ typically fall within **1–100** and consistently yield strong results.
>
> (2) **Fast sweep using a small validation set (resource- or time-constrained case).** When resources are limited, we recommend sweeping using only **50–100 validation samples**. This process is extremely fast (e.g., **~5 minutes** on an H100 GPU for 100 samples for the COPA task). Results using this small-set search are shown in below Table R4. It can be observed that even with only very few samples for hyperparameter selection, our proposed method still significantly outperforms the baseline methods and achieves results comparable to the full search.
>
> **Table R4. Results on LLaMA2-7B-Chat**
>
> | Method   | Avg. Acc. (5 tasks) | Detox | BPO  |
> |---|---|---|---|
> | Vanilla   | 56.01   | –      | –    |
> | SADI  | 59.49   | 86.32 | 13.50 |
> | AUSteer (100-sample search)   | 61.03   | 88.49 | 22.00 |
> | AUSteer (Full search)     | 61.34     | 89.24 | 22.00 |
>
> (3) **Hyperparameter sensitivity of different tasks**.  The heatmaps in Figure 10 in our paper provide a clear view of how the two hyperparameters—$k$ (number of steered AUs) and $\alpha$ (steering strength)—affect AUSteer across COPA, MAWPS, and BPO. Several consistent observations emerge. (i) **Task-specificity.** The optimal $\alpha, k$ values differ across tasks: COPA prefers moderately large values, MAWPS shows uniformly strong performance across almost all settings, and BPO benefits from moderate $\alpha$. This confirms that hyperparameters are task-dependent, consistent with prior activation-steering work [8-11]. (ii) **Robustness within each task.** Despite task-specific optima, performance varies very little within each task’s neighborhood of good settings. For COPA, accuracy fluctuates within ~1.5% across a broad range. This demonstrates that AUSteer is not sensitive to small changes in hyperparameters.
>
>
> In summary, although the hyperparameters remain robust within an individual task, task-specific hyperparameters are still a common challenge in activation steering. The standard solution used widely in existing studies is to perform a sweep [8-11]. To further reduce cost, we show that sweeping on a very small validation set is both **efficient** and **highly effective**, while still outperforming strong baselines. We will explore more principled approaches to reducing task-dependent hyperparameter sensitivity in future work.

---

> ### Author Response · Authors · 2025-11-19
> **Response N/N**
>
> [**W5**] One wonders what is the runtime overhead for AU localization and steering per sample compared to block-level methods like SADI?
>
> [**A5**] Thank you for your question. We compare the overhead for both the preparation stage (computing on contrastive samples, including activation extraction, localization, and steering-vector estimation) and the inference (steering) stage. We find that our method requires less running time than baseline methods, while achieving stronger steering performance. More details are provided as follows.
>
> We examine the computational overhead of our method and all baselines at each stage of the method. In the preparation phase, AUSteer extracts activations from contrastive pairs to compute activation momentum, whereas baseline methods usually require component localization and steering-vector estimation. During inference, each method applies its corresponding intervention, and we compare the resulting overhead across methods.
>
> Table R5 compares the computation cost of AUSteer with ITI and SADI across six metrics: 1) GPU memory for contrastive samples of all tasks, 2) total runtime on all contrastive samples, 3) GPU memory during inference, 4) inference time over seven tasks, 5) latency, and 6) latency stability (std from five repeated trials). The backbone LLM is Gemma2-9B-it (batch size = 1, GPU = NVIDIA H100).
>
> AUSteer requires **only ~15 minutes** to compute activation momentum, **no additional GPU memory** beyond inference, and exhibits **lower overhead** than ITI and SADI while achieving **better performance**, demonstrating its computational efficiency. During inference (steering), AUSteer also requires slightly less time than the baseline methods, further demonstrating its efficiency in runtime overhead.
>
> **Table R5: Overhead Comparison**
>
> | Method    | GPU Memory (on contrastive samples) | Required Time (on contrastive samples) | GPU Memory (Inference) | Inference Time | Latency  | Stability   |
> |---------|----------|-------|-----|-------|------|------|
> | Vanilla LLM      | -     | –    |  18 GB   | 53 min 12 sec  | 0.45 s/sample  | Δ 0.005  |
> | ITI      | 18 GB  | 18 min 39 sec | 18 GB    | 59 min 29 sec  | 0.50 s/sample  | Δ 0.01  |
> | SADI     | 18 GB    | 14 min 41 sec  | 18 GB    | 55 min 05 sec  | 0.47 s/sample  | Δ 0.005    |
> | AUSteer (Ours)   | 18 GB  | 14 min 41 sec | 18 GB      | 54 min 41 sec  | 0.46 s/sample  | Δ 0.007   |
>
> -----
>
> **Thank you again for taking the time to review our paper. All of the discussed clarifications have been incorporated into the revised version. We look forward to your feedback.**

---

### Official Review · Reviewer_61go · 2025-11-07

**Soundness:** 3
**Presentation:** 3
**Contribution:** 3
**Rating:** 6
**Confidence:** 3

**Summary:**

In this paper, they present AUSteer, a more fine-grained activation steering technique, to control LLM's behavior during the inference time. First, they recognize the heterogeneity in block activation and explain this through comprehensive experiments. Inspired by these experiments, they developed a more fine-grained activation steering algorithm. In detail, first, use activation momentum to recognize the important atomic unit on the target tasks. Then, steer these atomic units' activation adaptively. They did comperihensive experiments to evaluate AUSteer. And the results are convincing.

**Strengths:**

1. They first use two sections to recognize and interpret the heterogeneity in block activation, which gives insight and inspiration for AUSteer.
2. The method is natural and effective.
3. The experiments are comprehensive, spanning three LLMs with different architectures and three different tasks.

**Weaknesses:**

1. The biggest model used is 27B. Evaluating AUSteer on bigger models, e.g., 32B and 72B, and sparse models, e.g., MoE, even multi-modal models would be better.
2. The optimal hyperparameters \alpha and k are task-specific; how to set the hyperparameters for every tasks? And what is the hyperparameters used in Table. 1?

**Questions:**

See weakness.

---

> ### Author Response · Authors · 2025-11-19
> **Response 1/2**
>
> Dear Reviewer 61go,
>
> Thank you for taking the time to review our paper and for highlighting the **insightful recognition of heterogeneity in block activations**, as well as describing our method as **natural**, **effective**, and supported by **comprehensive experiments**. Your concerns primarily focus on the **evaluation on larger (30B–70B) and sparse LLMs**, as well as the **detailed analysis and clarification of hyperparameters**. We have carefully addressed all of these points and incorporated the corresponding revisions into the updated paper. We are pleased to see that, with your suggestions, the quality of the paper has been significantly improved. Our responses **[R]** to the raised weaknesses **[W]** are summarized below.
>
>
> [**W1**] The biggest model used is 27B. Evaluating AUSteer on bigger models, e.g., 32B and 72B, and sparse models, e.g., MoE, even multi-modal models would be better.
>
> [**A1**] Thank you for your suggestions. We conducted additional experiments on 3 commonsense reasoning tasks using larger LLMs ranging from 30B to 70B parameters, covering **general**, **sparse**, and **heavily quantized** architectures. Across all settings, AUSteer consistently improves performance over the vanilla models, demonstrating strong scalability. These results have been added to Appendix G of the revised paper. The details are as follows.
>
> To further verify the generalizability and scalability of AUSteer, we evaluate it on two representative large models with diverse structures: (1) **Qwen3-30B-A3B**, a 30B-scale **sparse MoE** model; and (2) **Llama-3.3-70B-Instruct**, where we use the **4-bit quantized** version to enable evaluation on a consumer GPU and to test AUSteer’s compatibility with **heavily quantized LLMs**. The results are shown in Table R1 below. In most cases, AUSteer improves performance by **1%–3%**, confirming its effectiveness and scalability across larger, structurally diverse and heavily quantized LLMs.
>
> **Table R1: More results of diverse and larger LLMs.**
>
> | Model                 | Method   | BoolQ | COPA | WinoG |
> |-----------------------|----------|-------|------|-------|
> | **Qwen3-30B-A3B**     | Vanilla  | 86.82 | 93.4 | 65.98 |
> |                       | AUSteer  | 88.69 | 97.8 | 67.17 |
> | **Llama-3.3-70B-Instruct** | Vanilla  | 89.54 | 98.6 | 78.14 |
> |                       | AUSteer  | 90.67 | 99.2 | 79.95 |

---

> ### Author Response · Authors · 2025-11-19
> **Response 2/2**
>
> [**W2**] The optimal hyperparameters \alpha and k are task-specific; how to set the hyperparameters for every tasks? And what is the hyperparameters used in Table. 1?
>
> [**A2**]  Thank you for your question. To set the hyperparameters for each task, we provide two solutions: (1) under sufficient computing resources, we perform a full hyperparameter sweep, which is consistent with previous studies [1,2,3,4]; and (2) in computing-constrained scenarios, we recommend using a very small validation set to conduct a quick hyperparameter sweep. In addition, (3) the optimal hyperparameters used in our experiments are reported in Tables R3a and R3b below. We explain these three points in detail as follows. These discussions have been added to Appendix C of our revised paper.
>
> - **General hyperparameter sweep (resource-sufficient case). Task-specific hyperparameters are still a common challenge in activation steering, and the standard solution used widely in existing studies is to perform a sweep [1-4].** Following [1–4], we perform a full hyperparameter sweep to empirically determine optimal $\alpha$ and $k$. We also run the same sweep for all baseline methods to ensure fair comparison in Table 1. Across tasks, both $\alpha$ and $k$ typically fall within **1–100** and consistently yield strong results.
>
> - **Fast sweep using a small validation set (resource- or time-constrained case).** When resources are limited, we recommend sweeping using only **50–100 validation samples**. This process is extremely fast (e.g., **~5 minutes** on an H100 GPU for 100 samples for the COPA task). Results using this small-set search are shown in below Table R2. It can be observed that even with only very few samples for hyperparameter selection, our proposed method still significantly outperforms the baseline methods and achieves results comparable to the full search.
>
> **Table R2. Results on LLaMA2-7B-Chat**
>
> | Method   | Avg. Acc. (5 tasks) | Detox | BPO  |
> |---|---|---|---|
> | Vanilla   | 56.01   | –      | –    |
> | SADI  | 59.49   | 86.32 | 13.50 |
> | AUSteer (100-sample search)   | 61.03   | 88.49 | 22.00 |
> | AUSteer (Full search)     | 61.34     | 89.24 | 22.00 |
>
> - **Hyperparameters used in Table 1.**
>   The optimal values of $\alpha$ and $k$ used for each task are reported in Tables R3a and R3b below. These values were obtained via full sweep, and the same process was applied to baseline methods for fairness. **The task-specific variation of hyperparameters aligns with observations from prior work** [1–4], indicating that different tasks may require different hyperparameter values. However, **for any given task, the hyperparameters are stable and robust**. For example, an shown in Figure 10 in our paper, for the COPA task, when $20 <= \alpha <= 50$ and $64 <= k <= 100$, the performance remains stable and varies within only 1.5%, while still significantly outperforming the baseline methods. For the MAWPS task, when $10 <= \alpha <= 50$ and $16 <= k <= 50$, the performance also varies within approximately 1.5%. Therefore, for each specific task, our method is hyperparameter-robust, and within the optimal region, it achieves comparable results with only small variations.
>
> **Table R3a. Optimal $\alpha$**
>
> | BoolQ | COPA | Winogrande | SVAMP | MAWPS | Detoxic. | BPO |
> |-------|------|---|-----|---|---|------|
> | 15    | 50   | 100    | 8      | 8  | 15        | 32   |
> | 50    | 50   | 100  | 100    | 50   | 8         | 10   |
> | 10    | 20   | 20   | 10     | 50  | 10  | 16   |
>
> **Table R3b. Optimal $k$**
>
> | BoolQ | COPA | Winogrande | SVAMP | MAWPS | Detoxic. | BPO |
> |-------|------|----|------|------|--------|----|
> | 100   | 16   | 2  | 50     | 80  | 16 | 16   |
> | 8     | 80   | 64  | 4  | 8      | 16   | 8    |
> | 100   | 8    | 100   | 100  | 2   | 8   | 10  |
>
> In summary, although the hyperparameters remain robust within an individual task, task-specific hyperparameters are still a common challenge in activation steering. The standard solution used widely in existing studies is to perform a sweep [1-4]. To further reduce cost, we show that sweeping on a very small validation set is both **efficient** and **highly effective**, while still outperforming strong baselines. We will explore more principled approaches to reducing task-dependent hyperparameter sensitivity in future work.
>
> References
>
> [1] Li, Kenneth, et al. *Inference-time intervention: Eliciting truthful answers from a language model.* NeurIPS 2023.
> [2] Rimsky, Nina, et al. *Steering Llama 2 via contrastive activation addition.* ACL2024.
> [3] Chen, Zhongzhi, et al. *Truth forest: Toward multi-scale truthfulness in LLMs through intervention without tuning.* AAAI 2024.
> [4] Wang, Weixuan, et al. *Semantics-adaptive activation intervention for llms via dynamic steering vectors.* ICLR 2025.
>
> ---
>
> **Thank you again for the insightful comments. We have incorporated the clarifications and discussions into the revised paper, and we look forward to your further feedback.**

---

### Author Response · Authors · 2025-11-19
**Summary**

We thank all reviewers for their careful evaluation of our work. The paper is recognized for its **identification of activation heterogeneity and natural, effective methodology** (Reviewer 61go), its **novel decomposition into atomic units and extensive experiments** (Reviewer mmMY), and its **intuitive, well-motivated fine-grained design with simplicity and interpretability** (Reviewer xSGu). We appreciate the consistent recognition of both the method’s conceptual novelty and empirical strength.

We have carefully addressed all concerns raised by the reviewers and summarize the revisions below:

1. **Scalability to larger and diverse LLMs:** We demonstrate AUSteer’s scalability on Qwen3-30B-A3B (sparse MoE) and Llama-3.3-70B-Instruct (4-bit quantized), confirming robust performance across architectures and model sizes.
2. **Theoretical and empirical justification of activation momentum:** We provide a clearer derivation and additional evidence showing how activation momentum contributes to discriminative causality and influences model outputs.
3. **Computation-overhead analysis:** We present detailed measurements showing that AUSteer achieves superior performance while requiring the least computational overhead among all baselines.
4. **Control-variant analysis:** We evaluate broader steering schemes (e.g., steering all AUs or large subsets) and show that effective steering must remain focused on task-relevant AUs, consistent with AUSteer’s design.
5. **Promotion vs. suppression:** Through theoretical reasoning and empirical results, we show that promoting beneficial AUs consistently outperforms suppressing unhelpful ones.
6. **Hyperparameter sensitivity and selection:** We provide expanded analyses and introduce a practical solution for hyperparameter selection in resource-constrained settings.
7. **Contrastive-sample construction:** Given that contrastive pairs are required by nearly all activation-steering methods, we offer a simpler, general, and empirically validated procedure for constructing them.

**While the main motivations, insights, proposed method, core experiments, and contributions remain unchanged, the added discussions and clarifications strengthen the paper by explaining several aspects more clearly and deeply. All updates have been incorporated into the revised version and highlighted in blue. Thank the reviewers again for their valuable feedback.**

---

### Meta-Review · Area_Chair_Uu59 · 2026-01-03

**Summary:**

Reviewers all agreed that the paper motivates its methods well, and the method is effective and simple. There are good experiments that have only been improved in the rebuttal period with larger models and runtimes.

A key concern was runtime and efficiency, which was claimed but not shown in the paper. Authors have rebutted this well by adding runtimes and running on larger models. I recommend that the runtimes table is included in the main paper itself.

Additional experiments regarding hyperparameter selection and sensitivity also improve the paper.

Reviewer's mmMY's concern about connecting activation momentum to discriminative causality was not sufficiently addressed in my opinion. The authors wrote some theoretical justification but I do not agree with it: first, there is a Taylor approximation (which can be quite a big approximation), and second, the gradient do/dx_i depends on x_i, meaning cannot make claims across different x_i, as the authors try to do.

**Reviewer Concerns:**

Please see summary: key reviewer concerns about efficiency and hyperparameters were well-addressed in the rebuttal. However, a concern about linking to causality was not and remains a concern. Additionally, I did not find the authors' rebuttal to reviewer xSGu's concern about comparison with a variant with all AU sufficient (but this is a minor point): I do not understand why this performs well than choosing a few AUs, as surely the method can learn to only use the relevant ones and set the weights of the others such that they are not used.

**Reviewer Scores:**

I expect 61go would either keep at 6 or maybe increase. I expect mmMY to either keep at 4 or increase to 6. I expect xSGu to increase to 6.

---

### Decision · Program_Chairs · 2026-01-26

Accept (Poster)